# Molecular basis for multimerization in the activation of the epidermal growth factor receptor

Yongjian Huang[1,2,3,4†], Shashank Bharill[1†], Deepti Karandur[1,2,3], Sean M Peterson[1,2,3], Morgan Marita[5], Xiaojun Shi[5], Megan J Kaliszewski[5], Adam W Smith[5*], Ehud Y Isacoff[1,2,4,6,7*], John Kuriyan[1,2,3,4,6,8*]

[1]Department of Molecular and Cell Biology, University of California, Berkeley, Berkeley, United States; [2]California Institute for Quantitative Biosciences, University of California, Berkeley, Berkeley, United States; [3]Howard Hughes Medical Institute, University of California, Berkeley, Berkeley, United States; [4]Biophysics Graduate Group, University of California, Berkeley, Berkeley, United States; [5]Department of Chemistry, University of Akron, Akron, United States; [6]Physical Biosciences Division, Lawrence Berkeley National Laboratory, Berkeley, United States; [7]Helen Wills Neuroscience Institute, University of California, Berkeley, Berkeley, United States; [8]Department of Chemistry, University of California, Berkeley, Berkeley, United States

*For correspondence: asmith5@uakron.edu (AWS); ehud@berkeley.edu (EYI); kuriyan@berkeley.edu (JK)

†These authors contributed equally to this work

**Abstract** The epidermal growth factor receptor (EGFR) is activated by dimerization, but activation also generates higher-order multimers, whose nature and function are poorly understood. We have characterized ligand-induced dimerization and multimerization of EGFR using single-molecule analysis, and show that multimerization can be blocked by mutations in a specific region of Domain IV of the extracellular module. These mutations reduce autophosphorylation of the C-terminal tail of EGFR and attenuate phosphorylation of phosphatidyl inositol 3-kinase, which is recruited by EGFR. The catalytic activity of EGFR is switched on through allosteric activation of one kinase domain by another, and we show that if this is restricted to dimers, then sites in the tail that are proximal to the kinase domain are phosphorylated in only one subunit. We propose a structural model for EGFR multimerization through self-association of ligand-bound dimers, in which the majority of kinase domains are activated cooperatively, thereby boosting tail phosphorylation.

## Introduction

The epidermal growth factor receptor (EGFR) is a receptor tyrosine kinase that couples the binding of extracellular ligands, such as EGF and transforming growth factor-a (TGF-α), to the initiation of intracellular signaling pathways that control cell growth and proliferation (*Kovacs et al., 2015a*; *Leahy, 2004*; *Lemmon et al., 2014*). Human EGFR is one of four closely related receptors that form homodimeric or heterodimeric combinations (*Yarden and Sliwkowski, 2001*). EGFR and one other member of the family (human epidermal growth factor receptor 4, HER4, also known as ErbB4) bind to specific ligands, and respond by activating their intracellular kinase domains. The two other members of the family, HER2 and HER3, are potent signaling receptors in combination, with HER3 switching on the kinase activity of HER2 in response to ligand binding (*Citri et al., 2003*; *Sliwkowski et al., 1994*). Aberrant signaling from EGFR family members underlies the onset of

many cancers, and there is intense interest in understanding how these receptors are regulated (*Lynch et al., 2004*; *Paez et al., 2004*; *Pao et al., 2004*; *Slamon et al., 2001*).

The canonical view of EGFR activation considers the monomeric receptor to convert to a dimeric form upon the addition of EGF (*Lemmon et al., 1997*; *Schlessinger, 2002*; *Yarden and Schlessinger, 1987a*; *1987b*) (*Figure 1A*). Dimerization activates the intracellular kinase domains of the receptor, resulting in autophosphorylation of a ∼230 residue C-terminal tail and the initiation of downstream signaling (*Margolis et al., 1989*). The extracellular module of EGFR consists of four domains, denoted Domains I, II, III and IV (*Figure 1A and B*). Domains I and III form the ligand-binding site of the receptor (*Garrett et al., 2002*; *Lu et al., 2010*; *Ogiso et al., 2002*). Domain II, which bridges the ligand-binding domains, contains a 'dimerization arm' that interacts with the corresponding element in the other subunit in a dimer (*Dawson et al., 2005*). Domains I, II and III form a compact unit that we refer to as the 'head' of the extracellular module, of which Domain IV forms an elongated 'leg'. In the absence of ligand, the extracellular module adopts a tethered and autoinhibited conformation, in which the dimerization arm is buried between the head and the leg (*Ferguson et al., 2003*). Ligand binding converts the receptor to a straightened form, releasing the dimerization arm, which then interacts to form a back-to-back dimer in which ligands do not participate directly in the inter-subunit interfaces.

On the intracellular side, activation of the receptor involves the formation of an asymmetric dimer of kinase domains, in which one kinase serves as a cyclin-like allosteric activator of the other (*Jura et al., 2009a*; *2011*; *Red Brewer et al., 2009*; *Zhang et al., 2006*). This allosteric interaction involves the C-lobe of one kinase (the 'activator') forming a tight interface with the N-lobe of another (the 'receiver'), which stabilizes the receiver in an active conformation (*Figure 1A*). The transmembrane helices, which connect the extracellular module to the intracellular module, also dimerize (*Mendrola et al., 2002*; *Mineev et al., 2010*), and this dimer is coupled to the asymmetric dimer of kinase domains (*Arkhipov et al., 2013a*; *Endres et al., 2013*; *Jura et al., 2009a*). This model for EGFR dimerization explains how a variety of ligands can activate the receptor, how heterodimeric combinations of the four EGFR family members can form, and how the catalytically impaired member of this family, HER3, can activate HER2 (*Endres et al., 2014*; *Jura et al., 2009b*; *Kovacs et al., 2015a*; *Shi et al., 2010*).

In their seminal analysis of the activation of EGFR by EGF, Yarden and Schlessinger showed that the addition of EGF to EGFR results in the formation of dimers or higher-order multimers in cells (*Yarden and Schlessinger, 1987a*; *1987b*). Although subsequent discussion of the activation of EGFR has focused on a monomer to dimer transition, Clayton, Burgess and colleagues concluded that EGFR forms tetramers upon activation, and that the tetramers are essential for signaling (*Clayton et al., 2005*; *2008*; *Kozer et al., 2013*). Single-molecule tracking studies of EGFR in live cells show that EGFR forms large clusters after activation (*Chung et al., 2010*). Thus, higher-order multimerization and clustering are important correlates of EGFR activation, but the structural details and functional consequences of larger assemblies are not clear.

We now present an analysis of the stoichiometry of EGFR complexes before and after activation, using stepwise photobleaching of fluorescently-tagged proteins expressed in Xenopus oocytes (*Chen et al., 2015*; *Ulbrich and Isacoff, 2007*). We find that EGFR exists as a monomer prior to ligand binding. Addition of EGF results predominantly in higher-order multimers, rather than just dimers. We have identified, through modeling and mutagenesis, a region within Domain IV of the extracellular module that is important for multimerization of the receptor, but is not required for dimerization. These data suggest that EGFR multimers are comprised of EGFR dimers that self-associate through Domain IV to form higher-order structures.

We also used pulsed-interleaved excitation fluorescence cross-correlation spectroscopy (PIE-FCCS) (*Müller et al., 2005*; *Smith, 2015*) to study human EGFR transfected into mammalian COS-7 cells. We show that EGF addition to EGFR in COS-7 cells results in the formation of multimers, rather than just dimers. Importantly, introduction of a Domain IV mutation that blocks multimerization of EGFR in the Xenopus oocyte assay prevents the formation of multimers in mammalian cells, with the response to EGF of the mutant EGFR being limited to dimer formation.

We tested the importance of multimer formation for EGFR activity in a human cell line (HEK-293). The mutations in Domain IV that reduce multimerization also reduced phosphorylation of Tyr 992, a site in the C-terminal tail that is proximal to the kinase domain, and Tyr 1173, a distal site. These mutations attenuate the phosphorylation of phosphatidyl inositol 3-kinase (PI3K), which binds to Tyr

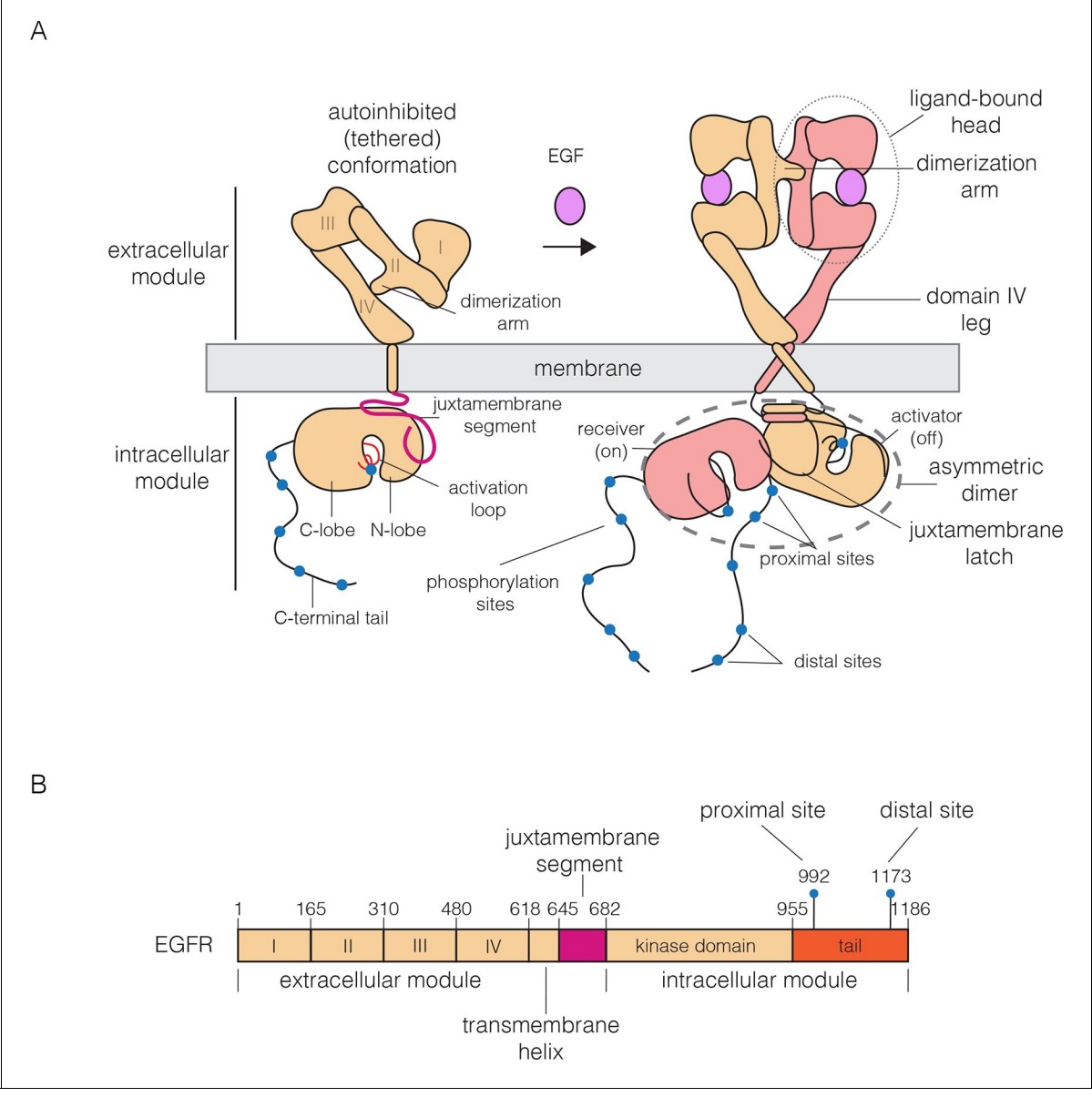

**Figure 1.** Schematic diagrams of EGFR structure. (**A**) The canonical model for the monomer to dimer transition in EGFR that is triggered by ligand binding (*Kovacs et al., 2015a*; *Lemmon et al., 2014*). (**B**) Domain boundaries in human EGFR. The residue numbers do not include the 24 residue signal sequence.

992 (*Jones et al., 2006*). In contrast, the phosphorylation of extracellular signal-regulated kinase (ERK), which is activated by distal phosphorylation of EGFR (*Batzer et al., 1994*), is not affected by these mutations. Our data suggest that this differential effect on proximal phosphorylation may arise from the structure of the asymmetric dimer of kinase domains, in which the proximal sites on the activator kinase cannot reach the active site of the receiver kinase, and would therefore be poorly phosphorylated in isolated dimers. We speculate that multimerization results in the formation of chains of kinase domains, in which the majority of kinase domains would be stabilized in the active form, allowing stronger phosphorylation in general, and more complete phosphorylation of proximal sites *in cis*.

We have constructed a model for a ligand-bound EGFR tetramer, using several criteria that emerged from our experiments. This model generates a spacing between dimers of the extracellular domain that is consistent with linkage to a head-to-tail chain of kinase domains, which serve as

receivers and activators for each other. The tetramer constructed in this way is open-ended, and can potentially engage additional dimers to extend the chain.

## Results and discussion

### Analysis of the stoichiometry of human EGFR by expression in Xenopus oocytes

We measured EGFR stoichiometry by transient transfection in Xenopus oocytes, which provides a convenient experimental system for controlled and reliable protein expression at very low levels (*Ulbrich and Isacoff, 2007*). It has been shown previously that human EGFR reconstituted into Xenopus oocytes is fully functional, and that the oocytes lack endogenous EGFR (*Opresko and Wiley, 1990*). For the Xenopus assay we made EGFR constructs that are fused at the C-terminus to EGFP, a monomeric variant of GFP that has enhanced fluorescence (*Yang et al., 1996*). We expressed EGFR-EGFP (we shall simply refer to this construct as EGFR, and to EGFP as GFP) in Xenopus oocytes, and used TIRF microscopy to image the animal pole of the oocyte (*Figure 2A*).

In a typical experiment, we observe well-separated spots of GFP fluorescence intensity corresponding to EGFR. The spots are stable in position, allowing their photobleaching properties to be studied readily. The surface density of the receptor is estimated to be ∼1 to ∼5 molecules per $\mu m^2$ (*Figure 2*). The effective local density of EGFR may be higher, because the membrane surface of the oocytes forms microvilli, within which the receptors are located (*Heinzmann and Höfler, 1994*; *Sonnleitner et al., 2002*). Nevertheless, the expression level of EGFR in these cells is very low when compared to that seen in typical mammalian-cell based experimental systems. For comparison, the surface densities of EGFR in the cancer cell lines A431, HeLa and A549 are ∼600, ∼300 and ∼100 molecules per $\mu m^2$, respectively (*Zhang et al., 2015*).

### EGFR is predominantly monomeric in the absence of ligand

To monitor photobleaching, a 13 μm x 13 μm field of view was illuminated in the TIRF mode by a laser and movies of 500–800 frames were recorded at a frame rate of 20 Hz. Frames from these movies were analyzed to measure the time-dependent decrease in the fluorescence intensities of individual spots, as shown in *Figure 2*. Analysis was restricted to spots that showed no overlap with nearby spots. In the absence of ligand, most spots undergo photobleaching in a single step, although some are observed to decay in two steps (*Figure 2B*). We do not see any spots with more than two steps in the photobleaching trace. We analyzed over 900 individual spots from ∼15 cells to generate a histogram of one-step and two-step photobleaching events, which shows that ∼94% of the spots show a single step-wise reduction in fluorescence intensity to baseline levels (*Figure 3A*). We conclude that the receptor is predominantly a monomer in the absence of ligand, at the expression level of this experiment.

We estimated the fraction of two-step photobleaching events that would occur due to random colocalization of non-interacting proteins. To do this, we expressed Claudin-16, a transmembrane protein that is known to be monomeric (*Gong et al., 2015*), at level comparable to that for EGFR in our experiments, and counted the number of one-step and two-step photobleaching events. For Claudin 16, ∼3% of the spots show two-step photobleaching. We subtracted this baseline fraction of two-step bleaching events from all the data shown in *Figure 3*. With this correction, the fraction of two-step photobleaching events for EGFR is ∼3% (*Figure 3A*). This is comparable to the level of ligand-independent preformed dimers reported in mammalian cells expressing EGFR at low levels (*Nagy et al., 2010*). Note that ∼25% to ∼35% of EGFP molecules are dark, due to incomplete maturation of the fluorophore, which results in fewer photobleaching steps than the number of subunits in a multimer (*Ulbrich and Isacoff, 2007*). Thus, two-step photobleaching may arise from dimers as well as a small population of higher-order multimers.

We deleted the dimerization arm of the extracellular module, involving residues 242–259 (the mutant is denoted Δ-arm). This resulted in an increase in the fraction of two-step photobleaching events, from ∼3% to ∼9% (*Figure 3B*). The increase in apparent dimerization upon deletion of the dimerization arm is unexpected, given that that this element forms a major part of the extracellular dimer interface. A possible explanation is that the dimerization arm is also important for maintaining the tethered and autoinhibited conformation of the extracellular module, and deletion of the

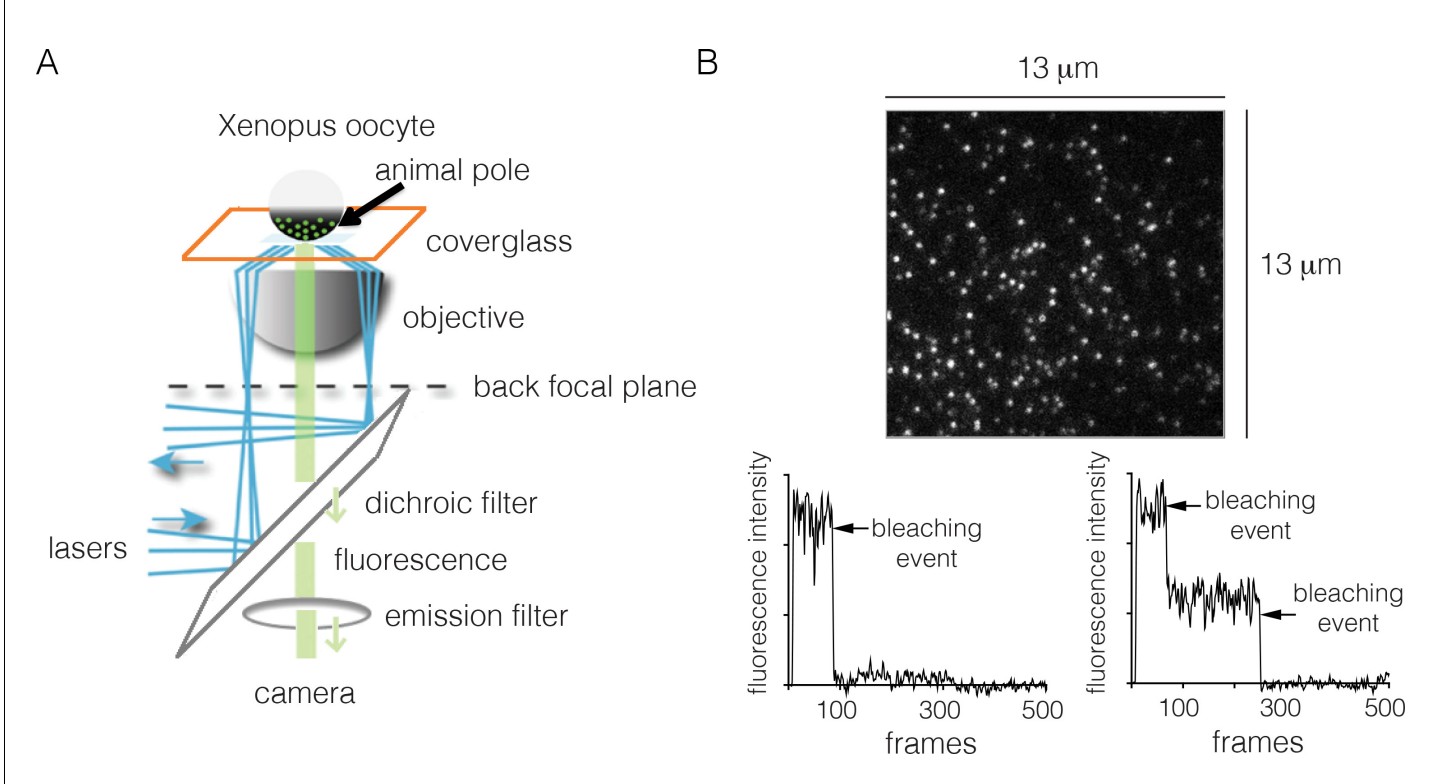

**Figure 2.** Monitoring the stoichiometry of human EGFR in Xenopus oocytes by photobleaching. (**A**) Schematic diagram illustrating the analysis of transiently-transfected EGFR by TIRF microscopy. A single oocyte is placed on a glass coverslip for observation and evanescent wave excitation is elicited by total internal reflection of the laser beam. The bright spots represent fluorescence from the EGFP fused to the C-terminus of EGFR. (**B**) Representative photobleaching traces of individual spots for EGFR without EGF. Image above shows a single image from a movie of the field of view. The two graphs show examples of one-step (left) and two-step (right) photobleaching. Frames collected at 20 Hz over 25 s.

dimerization arm is expected to release autoinhibition, and thereby facilitate dimerization (*Ferguson et al., 2003*). To test this idea we introduced mutations on the other side of the tethered interaction, which is in Domain IV rather than in the dimerization arm of Domain II. We introduced two point mutations (D563A, H566A) and also deleted a loop (residues 575 to 585) in Domain IV. A similar mutation in EGFR has been shown to abolish the tethering interaction (*Dawson et al., 2007*). This mutation, which we refer to as Δ-tether, has the dimerization arm intact, and it also increases the fraction of two-step bleaching events (*Figure 3C*). These results are consistent with the accepted view, which is that the autoinhibited conformation of the extracellular domain provides a steric block to dimerization (*Ferguson et al., 2003*). Once this block is released, the extended shape of the extracellular module permits dimerization even if the dimerization arm is deleted.

## Addition of EGF to EGFR generates dimers and higher-order multimers

We added EGF at 15 nM concentration to Xenopus oocytes expressing EGFR. The EGF was added to cells in suspension, before they were placed on the microscope slide, since the surface of the cell that interacts tightly with the glass is inaccessible to EGF. The EGF concentration is 8-fold higher than the measured IC-50 value of 1.9 nM for EGF binding to EGFR in a competition assay (*Jones et al., 1999*). We monitored photobleaching within 2 min of EGF addition, which minimizes the effects of receptor internalization. With the addition of EGF, we observed two-step photobleaching for some spots, as well as multistep photobleaching (three or more steps) for nearly half of the spots (*Figure 4A,B*). As can be seen by comparing *Figures 2B* and *4A*, the addition of EGF results in the appearance of some particularly bright spots that are also irregular in shape. We excluded such spots from the analysis.

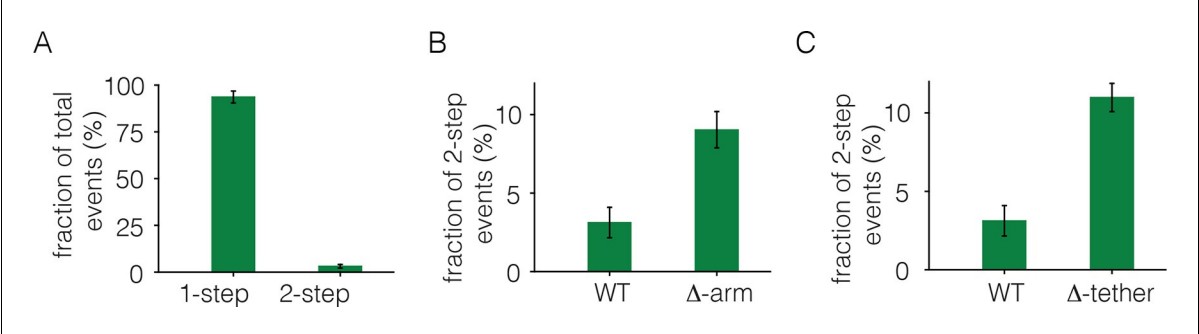

**Figure 3.** Analysis of photobleaching events for EGFR without EGF, in the Xenopus assay. (**A**) Histogram of one-step and two-step photobleaching events for wild-type EGFR in the absence of EGF. This histogram represents the number of counted events. For subsequent analysis, the fraction of two-step events (3%) seen for a monomer control protein (Claudin-16) in a similar experiment at a comparable density was subtracted from the fraction of two-step events seen for EGFR and EGFR variants. The fraction of two-step photobleaching events for wild-type EGFR (WT) is compared to that for mutants. (**B**) Δ-arm, deletion of the dimerization arm, residues 242 to 259. (**C**) Δ-tether, mutation in Domain IV that breaks the autoinhibitory interaction, (D563A, H566A and deletion of residues 575 to 585).

We grouped all spots for which there are more than two-steps in the photobleaching trace into one category, referred to as multistep bleaching. Roughly 50% of the spots show multistep bleaching, ~25% show two-step bleaching and ~25% show one-step bleaching (*Figure 4B*). We analyzed photobleaching data for an EGFR variant in which the catalytic base is mutated (D813N). This kinase-dead variant of EGFR exhibits a similar degree of multimerization to wild-type EGFR, demonstrating that the formation of multimers does not require signaling from active kinases (*Figure 4C*).

We repeated the step-wise photobleaching experiments using TGF-α instead of EGF, and found a similar distribution of one-step, two-step and multistep photobleaching events (*Figure 4D*). Although the structure of EGF is very similar to that of TGF-α, the residues that are surface exposed when EGF is bound to EGFR are not conserved in TGF-α (*Garrett et al., 2002*; *Ogiso et al., 2002*). We conclude that neither EGF nor TGF-α is likely to be at the interfaces between protomers in the multimers. The binding of EGF to EGFR exhibits negative cooperativity, which has been interpreted in terms of half-site binding of EGFR to EGF (i.e., at a stoichiometric ratio of 2:1) at low ligand concentrations (*Alvarado et al., 2010*; *Arkhipov et al., 2013b*; *Liu et al., 2012*; *Pike, 2012*). We analyzed step-wise photobleaching events upon addition of 2 µM EGF, instead of the 15 nM used in most of the experiments reported here. This 133-fold increase in ligand concentration did not lead to a substantial difference in the relative population of multimers, indicating that the multimer interface does not involve empty ligand binding sites (*Figure 4E*).

## Analysis of the role of the dimer interface in EGFR multimerization

We made mutations that are expected, separately, to disrupt the dimerization of the extracellular module, the transmembrane helices, and the kinase domains. We deleted the dimerization arm in the extracellular module (Δ-arm) and found that the extent of multimerization was not affected substantially (*Figure 5*). Deletion of the dimerization arm has both positive and negative effects on the receptor dimerization, as noted earlier, which might account for the neutral effect of this deletion on multimerization. We disrupted both the N- and C-terminal dimerization interfaces of the transmembrane helix (*Arkhipov et al., 2013a*; *Endres et al., 2013*; *Fleishman et al., 2002*), by replacing Thr 624, Gly 625, Gly 628, Gly 629, Ala 637 and Gly 641 simultaneously by isoleucine (*Endres et al., 2013*). To disrupt the asymmetric dimer interface in kinase domains, we introduced two mutations (I682Q, V924R) (*Zhang et al., 2006*). Both sets of mutations resulted in substantial reduction in the extent of multimerization (*Figure 5*).

We wondered if the transmembrane helices alone could be responsible for multimerization, perhaps by forming separate cross-bridges through the N-terminal and C-terminal dimerization elements. To test this, we made a construct in which the transmembrane helix of EGFR was fused to mCherry at the N-terminus and EGFP at the C-terminus. This construct was studied by photobleaching, and it exhibited one-step and two-step photobleaching, with no evidence for multistep

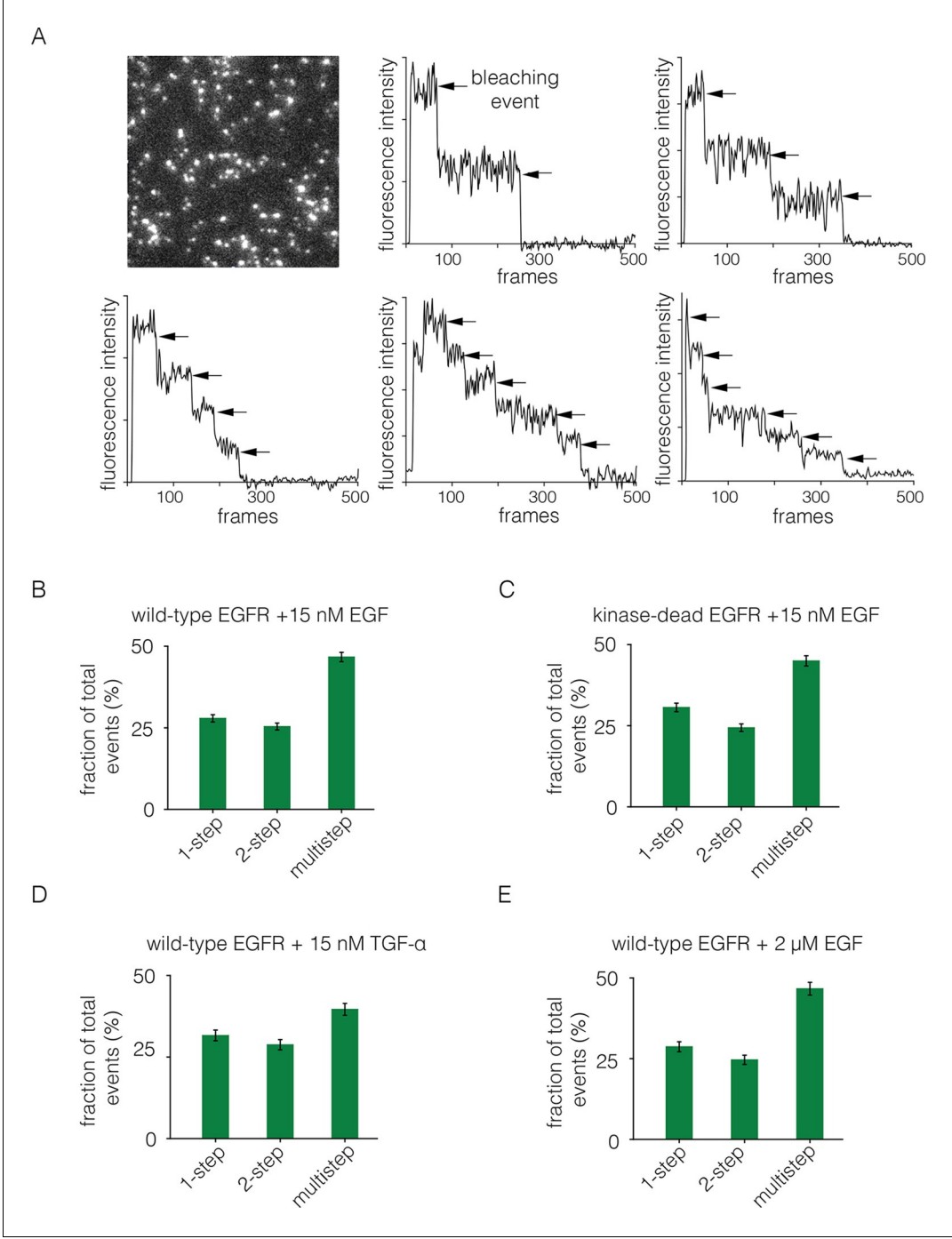

**Figure 4.** Analysis of photobleaching events for EGFR in the presence of EGF. (**A**) TIRF image of a Xenopus oocyte expressing EGFR, 2 mins after addition of 15 nM EGF. Five representative photobleaching traces are shown, in which two to six photobleaching events can be identified. Traces with more than two steps are denoted 'multistep'. (**B**) Histogram showing the fraction of one-step, two-step and multistep photobleaching events for wild-type EGFR in the presence of 15 nM EGF. (**C**) As in B, for a kinase-dead mutant of EGFR (D813N). (**D**) As in B, for TGF-α as the ligand, at a concentration of 15 nM. (**E**) As in B, but for a 133-fold higher concentration of EGF (2 μM).

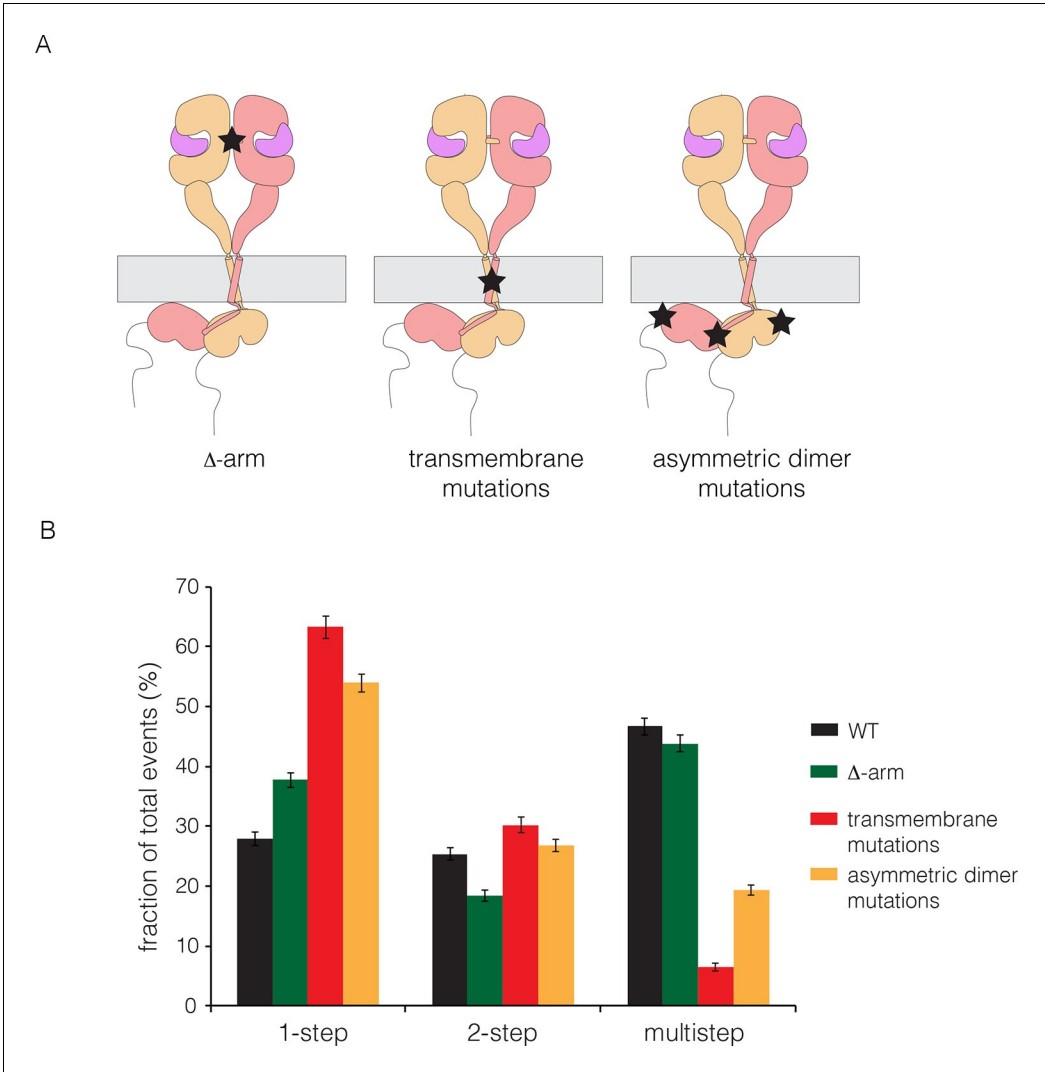

**Figure 5.** Effects on multimerization of mutations in EGFR that also affect dimerization. (**A**) Schematic diagram showing the locations of mutations with respect to the EGFR dimer. Δ-arm, deletion of the dimerization arm of the extracellular module. TM, mutations in the transmembrane helix (Thr 624, Gly 625, Gly 628, Gly 629, Ala 637 and Gly 641 are all replaced by isoleucine). Asymmetric dimer interface mutations combine the mutations I682Q in the N-lobe and V924R in the C-lobe of the kinase domain. (**B**) Histogram showing the fraction of one-step, two-step and multistep photobleaching events in Xenopus oocytes for wild-type and mutant EGFRs.

The following figure supplement is available for figure 5:

**Figure supplement 1.** Dimerization of the isolated transmembrane helix.

---

photobleaching (*Figure 5—figure supplement 1*). We conclude that the transmembrane helices do not form multimers by themselves at the low expression levels used in our experiments. An interesting aspect of these data is that the extent of dimerization for the isolated transmembrane helices (∼13%) is greater than that seen for full-length EGFR (∼3%) without EGF (compare *Figures 5—figure supplement 1* and *Figure 3A*). This is consistent with the idea that the tethered conformation of the ligand-free extracellular domain blocks receptor dimerization.

## Identification of residues in Domain IV that are necessary for EGFR multimerization

It is difficult to separate the structural requirements for multimerization from those necessary for dimerization. We wondered whether the extracellular domain has an interface that is required for multimerization, but not for dimerization or ligand binding. Packing interactions in crystal structures of the extracellular domains of EGFR family members do not provide obvious clues concerning such an interface. We therefore turned to computational docking to see if there are ways in which the extracellular domains might self-associate.

We first ran the docking program ClusPro, which emphasizes shape complementarity between protomers in generating models for protein complexes (*Kozakov et al., 2013*), using the structures of the entire dimeric extracellular module (PDB code: 3NJP) or a dimer containing just the ligand-bound head (PDB code: 1MOX), with no constraints. This led to docking solutions that were either physically implausible (i.e., the two dimers of the extracellular module could not be connected to the same membrane), or that had EGF at the interface between dimers. Since our experiments made it unlikely that EGF bridges EGFR dimers in a multimers, we concluded that ClusPro is unable to provide plausible models for self-interaction, which we expect to be very weak in the absence of organization on the membrane.

We then ran ClusPro using a single copy of the structure of the Domain IV leg alone. Although this did not result in one clear solution, we were intrigued by a family of solutions in which one face of Domain IV was packed upon itself in a dimer (*Figure 6A*). These solutions have a surface containing several hydrophobic residues at the interface between two Domain IV molecules (*Figure 6B*). This surface is not involved in formation of the EGF-bound EGFR dimer, and it is distal to the surface of Domain IV that forms the tethering interaction in autoinhibited ligand-free EGFR. This surface region therefore represents a potential interface for formation of a multimer of dimers.

There are three isoleucine residues in this surface region (residues 545, 556, 562) and one valine residue (Val 592) (*Figure 6B*, left). We first made two quadruple mutations, in which both I545 and I556 were replaced by either lysine or alanine, while I562 and V592 were mutated to arginine and glutamate, respectively. The variant harboring mutations I545K, I556K, I562R and V592E (denoted 'IIIV/KKRE') shows significant reduction of multimerization (*Figure 6C*). In contrast, the variant with I545A, I556A, I562R and V592E ('IIIV/AARE'), shows a similar extent of multimerization as wild-type EGFR (*Figure 6C*). We discuss below the effect of these mutations on the activity of the receptor. We found that the effect on EGFR activity of a double mutation (I545K and I556K; 'II/KK') is similar to that of the IIIV/KKRE quadruple mutation (not shown). We concluded that Ile 545 and Ile 556, which are further from the membrane and closer to Domain III, are important for multimerization, whereas the residues that are closer to the membrane, Ile 562 and Val 592, are not. We replaced three residues in the membrane-distal region of the predicted interaction surface, Val 526, Glu 527 and Asn 528, by glutamate, arginine and arginine, respectively ('VEN/ERR'; *Figure 6B*, right). In another variant we replaced Thr 548 and Asn 554 by argininines ('TN/RR'; *Figure 6B*, right). Both of these variants also show reduced multimerization (*Figure 6C*). Taken together, these data suggest that the residues in Domain IV shown in *Figure 6B*, spanning residues 526 to 556, are involved in multimerization of the receptor. We note that we have tested only a limited set of mutations, and that therefore our list of residues within a putative interfacial region is likely to be incomplete.

## Analysis of the stoichiometry of human EGFR in mammalian cells by PIE-FCCS

To assess the level of multimerization of various EGFR constructs in mammalian cells, we used two-color pulsed-interleaved excitation fluorescence cross-correlation spectroscopy (PIE-FCCS), in which a pair of lasers alternately excites EGFP and mCherry with subnanosecond pulses (*Muller et al., 2005*). Diffusion of individual molecules in and out of the diffraction-limited laser focus gives rise to fluorescence intensity fluctuations, from which we calculate a cross-correlation value that is proportional to the fraction of co-diffusing molecules (*Digman and Gratton, 2009*; *Marita et al., 2015*). Although PIE-FCCS cannot define precise stoichiometry, the comparison of cross-correlation values for different species allows the determination of the relative sizes of co-diffusing species. For these experiments, EGFR was expressed by transient transfection in COS7 cells, and data were measured using dozens of cells for each construct studied. The surface density of EGFR in these experiments

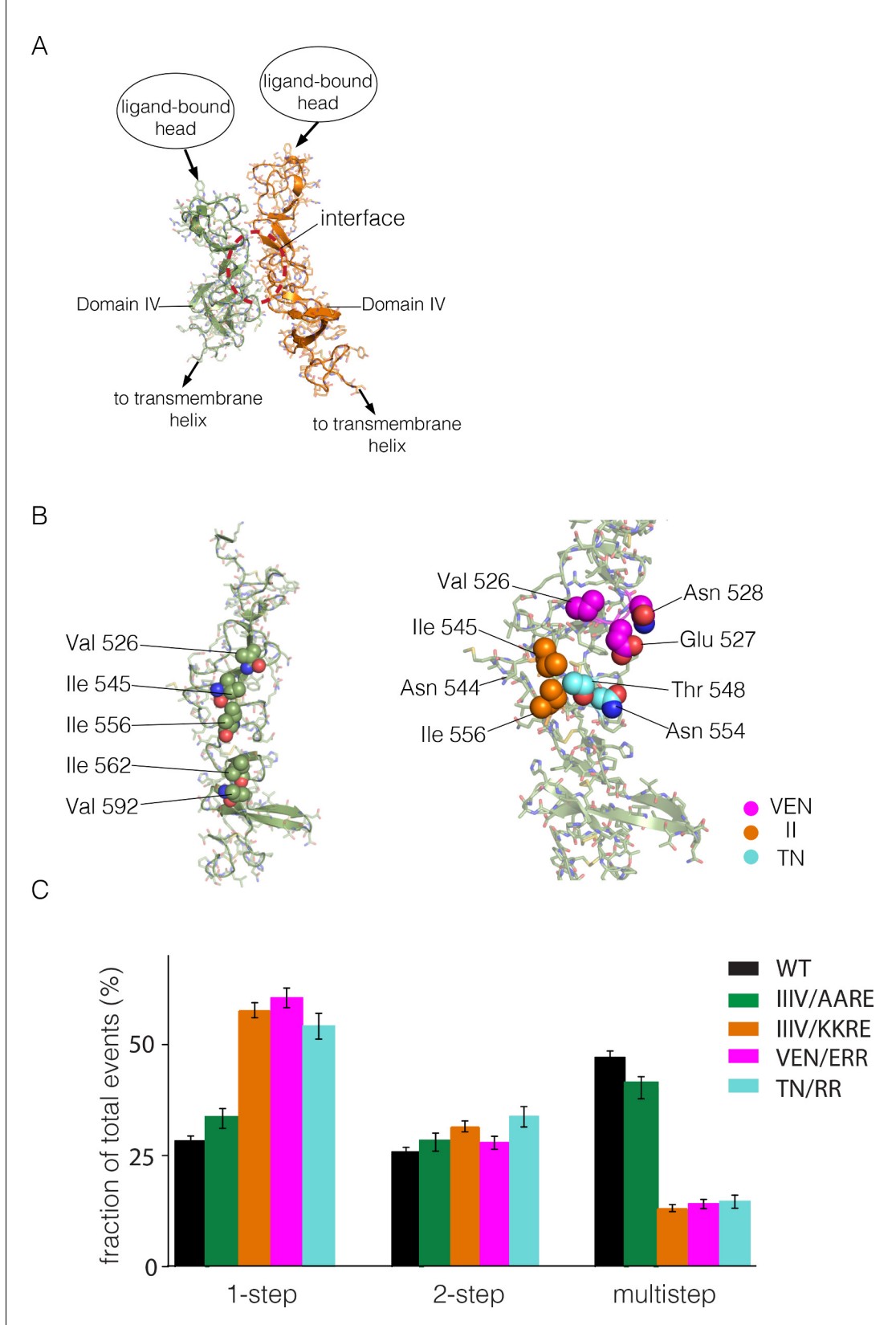

**Figure 6.** Identification of a possible interface between the Domain IV legs of two EGFR molecules. (**A**) A representative dimer generated by ClusPro from a docking calculation done using a single isolated Domain IV. The interface between the two domains is not involved in dimerization, and is on

*Figure 6 continued on next page*

*Figure 6 continued*

the opposite face of the domain with respect to the interface involved in autoinhibitory tethering. (**B**) Residues on Domain IV that were mutated to examine their effects on multimerization and activity. (**C**) Histogram of the fractions of one-step, two-step and multistep photobleaching events in Xenopus oocytes for wild-type and mutant EGFR. IIIV/AARE: I545A, I556A, I562R and V592E. IIIV/KKRE: I545K, I556K, I562R and V592E. VEN/ERR: V526E, E527R and N528R. TN/RR: T548R and N554R.

(100 to 1000 molecules per $\mu m^2$) is roughly 200-fold higher than in the Xenopus oocyte experiments (1 to 5 molecules per $\mu m^2$).

As a calibration for the expected cross-correlation values for a monomer at the membrane, we used a coexpression of EGFP and mCherry tagged with the c-Src membrane localization tag ('monomer control', median value of fraction correlated, $f_c$ = 0.01) (*Figure 7A*), as described (*Endres et al., 2013*). As calibration for a dimer, we used EGFP and mCherry fused to GCN4, with the same membrane-localization tag ('dimer control', $f_c$ = 0.13; GCN4 is a strong dimer) (*Marita et al., 2015*). We compared these data to PIE-FCCS data for constructs in which full-length EGFR is fused to EGFP and mCherry (*Figure 7B*, see also *Figure 7—figure supplement 1*). In the absence of ligand, the cross-correlation values for full-length EGFR are similar to those obtained for the monomer control, $f_c$ = 0.01 (*Endres et al., 2013*). In the presence of EGF, the cross-correlation values are significantly larger than those obtained for the dimer control, $f_c$ = 0.19, indicating that EGFR forms multimers upon binding EGF. We then introduced the quadruple mutation that disrupted multimer formation in Xenopus oocytes ('IIIV/KKRE'). Without EGF, the IIIV/KKRE mutant showed no significant dimerization, $f_c$ = 0.01, similar to wild type EGFR. Stimulation with 15 nM EGF resulted in an increase in cross-correlation to $f_c$ = 0.11, which is significantly reduced compared to wild-type EGFR and comparable to that of the dimer control. Diffusion constants extracted from the data support the conclusion that the IIIV/KKRE mutation impairs the ability of EGFR to form higher-order multimers upon binding EGF (*Figure 7C*). We conclude that EGFR forms ligand-dependent multimers in mammalian cells and, as for the Xenopus oocyte assay, that the multimers can be disrupted by mutations in a specific region of Domain IV.

We note that the cross-correlation values we observed for EGFR in the presence and absence of EGF are similar to those reported in our earlier study (*Endres et al., 2013*). Our previous study, however, lacked a proper dimer control. In that study, we had assumed that a construct in which the intracellular module (ICM) of EGFR was fused to GCN4 (GCN4-ICM) corresponded to a dimer. The current data show that GCN4-ICM forms multimers, because the cross-correlation value for this construct, $f_c$ = 0.23, is significantly higher than for the dimer control, $f_c$ = 0.13, and slightly higher than for EGF-bound EGFR, = 0.19. We are uncertain of the nature of the multimers formed by the GCN4-ICM construct, because introduction of mutations at the asymmetric dimer interface (I682Q, V924R) does not disrupt these multimers.

## Functional role of multimerization in EGFR autophosphorylation and downstream signaling

The multimerization of EGFR has been reported before, and it has been noted that the formation of tetramers is correlated with the recruitment of the adapter protein GRB2 (*Clayton et al., 2005*; *2008*; *Kozer et al., 2013*; *2014*). Nevertheless, a functional role for ligand-dependent multimerization has yet to be established definitively. The mutations that we have identified in Domain IV of EGFR that block multimerization, but not dimerization, provide an opportunity to identify the effects of multimerization on EGFR function. To test the effect of these mutations on the activity of EGFR, we used transient transfection in mammalian cells (HEK293T), followed by measurement of phosphorylation on EGFR and downstream signaling effector proteins by FACS assay, as described (*Kovacs et al., 2015b*). We monitored one proximal site in the EGFR C-terminal tail, Tyr 992, and one distal site, Tyr 1173 (antibodies with minimal cross-reactivity are available for these sites, see [*Kovacs et al., 2015b*]).

We compared autophosphorylation of wild-type EGFR with that of three variants with mutations in Domain IV (II/KK; VEN/ERR; TN/RR; see *Figure 6B* for the positions of the mutated residues). *Figure 8* shows the phosphorylation levels for Tyr 992, the proximal site, and Tyr 1173, the distal site, for wild-type EGFR and the mutants at an intermediate level of expression in the FACS assay. Each

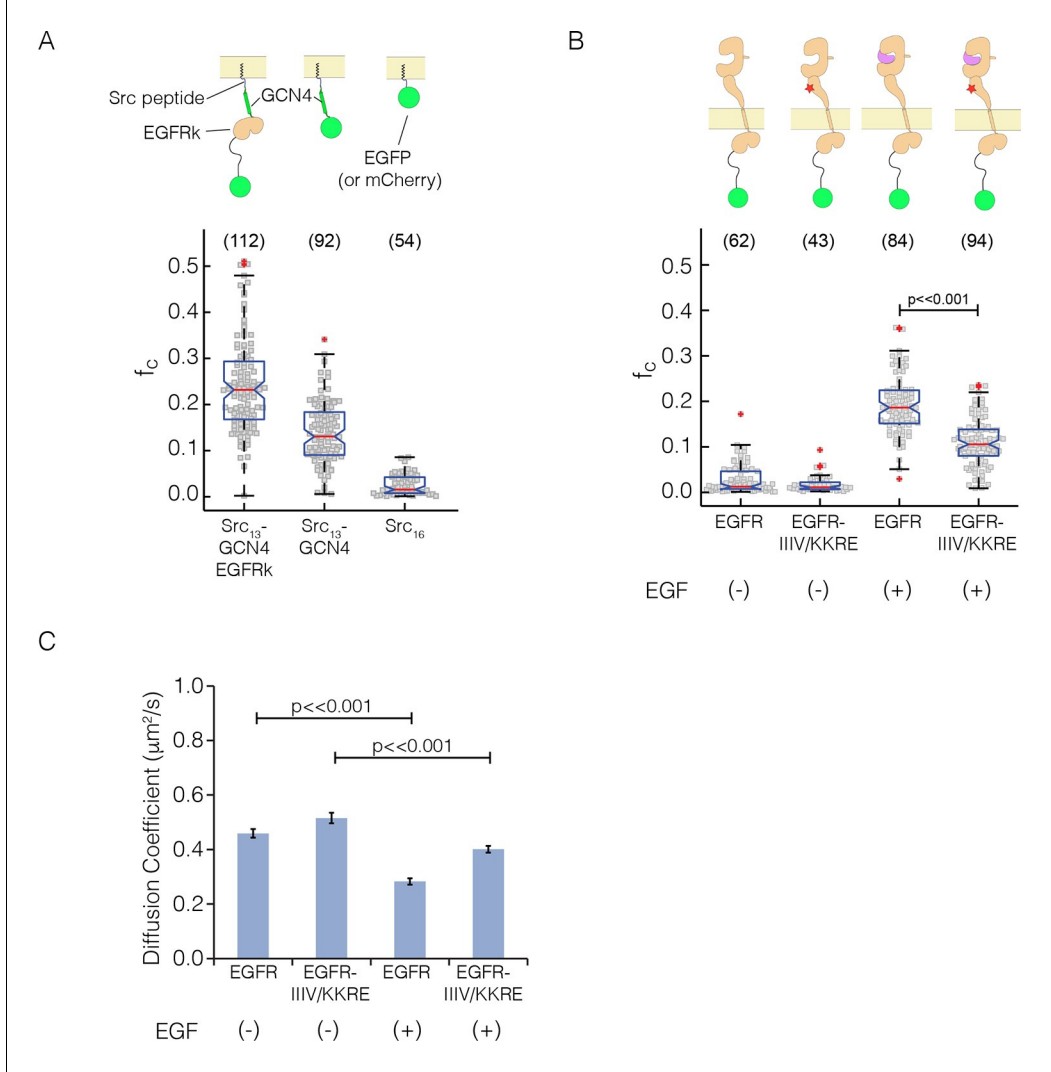

**Figure 7.** Pulsed-interleaved excitation fluorescence cross-correlation spectroscopy (PIE-FCCS) data for EGFR expressed in COS-7 cells. For each protein, EGFP and mCherry fusions are coexpressed via transient transfection, and the cross-correlation between them is calculated for individual cells and reported as a fraction correlated, $f_c$, indicated in the scatter plot. The red line represents the median value of the distribution, with the box enclosing the 25–75 percentile and the whiskers the entire distribution. Outliers are indicated by a red cross. (**A**) Data for control constructs, shown schematically at the top of the panel. These constructs are localized to the membrane by a thirteen residue N-terminal sequence from c-Src, and have either EGFP or mCherry fused to the C-terminus, as described (**Endres et al., 2013**). A multimer control, Src$_{13}$-GCN4-EGFRk, is a construct with the coiled-coil segment of GCN4 joined to the N-terminus of the kinase domain of EGFR. The dimer control, Src$_{13}$-GCN4, contains just the coiled-coil segment of GCN4 in addition to the membrane tag and the fluorescent protein. The monomer control, Src$_{16}$, contains only the membrane localization sequence and the flurescent protein. Note that the cross-correlation values for multimeric Src$_{13}$-GCN4-EGFRk are higher than that for the dimer control, as previously reported (**Marita et al., 2015**). (PIE-FCCS data for these controls were collected previously on the same instrument, and are reproduced here with permission.) (**B**) Relative cross-correlation values for full-length EGFR constructs, with and without the addition of EGF: EGFR, wild-type EGFR; and EGFR-IIIV/KKRE, an EGFR quadruple mutant (I545K, I556K, I562R and V592E). (**C**) Protein mobility from PIE-FCCS data. The mobility of each construct in the plasma membrane is directly measured with PIE-FCCS and converted to an effective diffusion coefficient for comparison. The diffusion coefficients of EGFR and EGFR-IIIV/KKRE before ligand stimulation are 0.46 ± 0.02 and 0.52 ± 0.02 μm$^2$/s respectively ( ± standard error). These values are consistent with monomeric, single pass transmembrane proteins in the plasma membrane (**Marita et al., 2015**). Upon stimulation with EGF, the diffusion coefficient of EGFR-IIIV/KKRE drops to 0.40 ± 0.01 μm$^2$/s, consistent with a monomer-dimer transition. For wild-
*Figure 7 continued on next page*

*Figure 7 continued*

type EGFR, EGF stimulation causes the diffusion coefficient to be reduced significantly more than for the mutant, to $0.28 \pm 0.01$ μm$^2$/s, consistent with the formation of multimers.

The following figure supplement is available for figure 7:

**Figure supplement 1.** Distributions of $f_c$ values for EGFR (+EGF), EGFR-IIIV/KKRE (+EGF) and the two positive controls, Src$_{13}$-GCN4-EGFRk and Src$_{13}$-GCN4.

of three mutants show reduced phosphorylation of both sites compared to wild-type. For example, for the II/KK mutant, phosphorylation of Tyr 992 and Tyr 1173 are reduced to ~35% and ~50% of the wild-type levels.

We verified that the surface expression of EGFR is not affected by mutations in Domain IV. To do this we made a construct in which an HA epitope tag is fused to the N-terminus of EGFR, with a fluorescent protein (Venus) fused to the C-terminus. This construct was transiently transfected into HEK293T cells, as for the activity assays shown above. An HA-specific antibody was used to visualize surface-exposed EGFR in individual cells using a fluorescence microscope, and the total expression of EGFR was monitored by detecting fluorescence of the Venus protein tag. As shown in *Figure 8— figure supplement 2*, there is no difference in the ratio of surface-exposed to total EGFR for wild type compared to the II/KK mutant, over the entire range of EGFR expression in the experiment.

We note that one of the glycosylation sites in EGFR, Asn 544 (*Zhen et al., 2003*), is located close to residues Ile 545 and Ile 556 (see *Figure 6B*). We wondered whether mutations in Domain IV could be affecting this glycosylation site, and perhaps altering multimerization and signaling in that way. We made two EGFR variants in which Asn 544 was replaced by either aspartate or glutamine. Both mutant receptors show no reduction in EGFR tail phosphorylation (data not shown).

We also monitored the phosphorylation of two proteins that are expected to be activated as a consequence of distal and proximal phosphorylation of the EGFR tail, respectively. Extracellular signal-related kinase (ERK) is activated by distal phosphorylation of EGFR, through the docking of the adapter proteins GRB2 and SHC (*Batzer et al., 1994*). Signaling from proximal sites in EGFR is less well characterized than for distal sites, but a comprehensive study of EGFR phosphopeptides binding to other proteins showed that the SH2 domains of the α and β isoforms of phosphatidyl inositol 3-kinase (PI3K), and that of phospholipase C-γ, bind with high affinity only to sites in the proximal segment of the EGFR tail (*Jones et al., 2006*). We monitored PI3K, and found that mutations in Domain IV result in attenuation of PI3K phosphorylation, to ~30% of the wild-type level (*Figure 8C*). In contrast, the phosphorylation of ERK by the MAP-kinase pathway is not affected by Domain IV mutations (*Figure 8D*).

To understand why mutations that reduce multimerization affect phosphorylation of the EGFR tail, we carried out co-transfection experiments using EGFR variants that are activator-impaired (i.e., a mutation prevents their kinase domains from taking the activator position in an asymmetric dimer) or receiver-impaired (a mutation prevents their kinase domains from taking the receiver position, but they can serve as activators; see *Figure 9A* for a schematic representation of these mutants). We showed previously that for such combinations the activator tail is phosphorylated slightly more efficiently than the receiver tail, but we could not distinguish between distal and proximal sites (*Kovacs et al., 2015b*).

We co-transfected cells with combinations of receiver-impaired and activator-impaired variants of EGFR in which the full-length tail is present on both kinases, but within which specific tyrosine residues are mutated in one tail but not the other. Phosphorylation of Tyr 1173 clearly occurs in both receiver and activator tails, with higher efficiency for the activator tail (*Figure 9A*), which is consistent with our previous study. For Tyr 992, in the proximal part of the tail, the phosphorylation level in the activator tail is much lower compared to that in the receiver tail (*Figure 9B*). Tyr 992, in the proximal part of the tail, can only access its own kinase domain in an asymmetric dimer (*Koland, 2014*). In a combination of receiver-impaired and activator-impaired EGFR molecules, the activator kinase domain is not itself activated, and so Tyr 992 in the activator tail cannot be phosphorylated efficiently. These results suggest that if phosphorylation occurred only within a dimer, then only half the

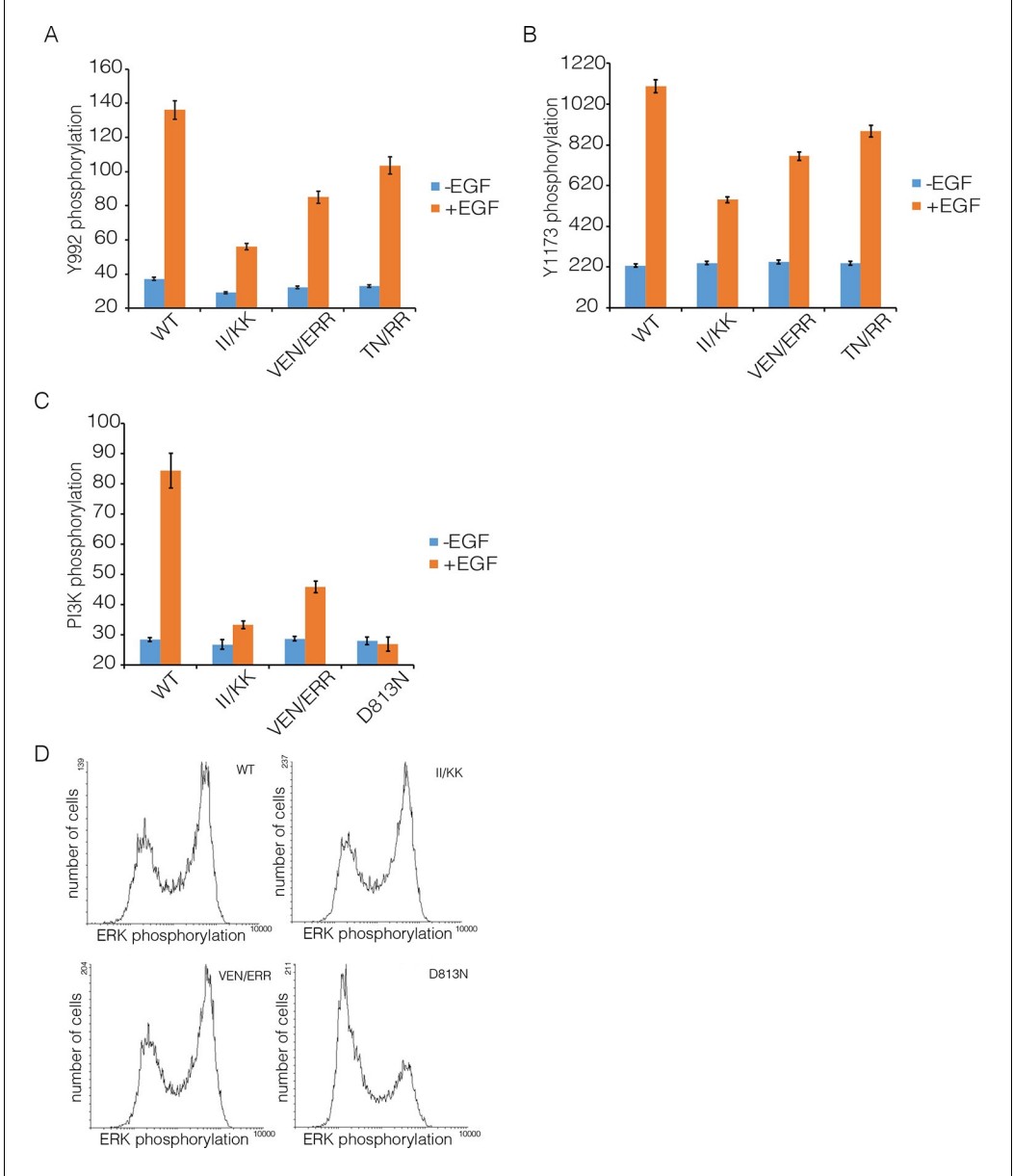

**Figure 8.** Effect of Domain IV mutations on phosphorylation of EGFR and two effector proteins. Wild-type (WT) or mutant EGFR was transfected into HEK293T cells, and phosphorylation was measured using FACS. The EGFR mutants are: II/KK (I545K, I556K); VEN/ERR (V526E, E527R and N528R); TN/RR (T548R and N554R). (**A**) Phosphorylation of a proximal site (Tyr 992) in EGFR. The bar graph shows the average phosphorylation level from cells expressing an intermediate level of EGFR, selected based on the whole range of EGFR expression in the FACS analysis, with (orange) and without (blue) the addition of EGF. (**B**) As in Panel A, for a distal site (Tyr 1173) in EGFR. (**C**) As in Panels A and B, for phosphorylation of PI3K. In addition to two mutations in Domain IV (II/KK and VEN/ERR), the phosphorylation levels for a kinase-dead EGFR variant (D813N) are shown. (**D**) Phosphorylation of ERK. The levels of phosphorylated ERK (pERK) show a bimodal distribution, with cells tending to have either low or high levels of pERK. The data are best represented as a histogram of cell numbers with different levels of pERK, as described (***Kovacs et al., 2015b***). Note that the response of two Domain IV mutants (II/KK and VEN/ERR) resembles that of the wild-type EGFR. For comparison, data for kinase-dead EGFR (D813N) show a reduced population of cells with high levels of pERK.

The following figure supplements are available for figure 8:

**Figure supplement 1.** Transmembrane helix mutations.

**Figure supplement 2.** Surface expression of wild-type EGFR and II/KK mutant.

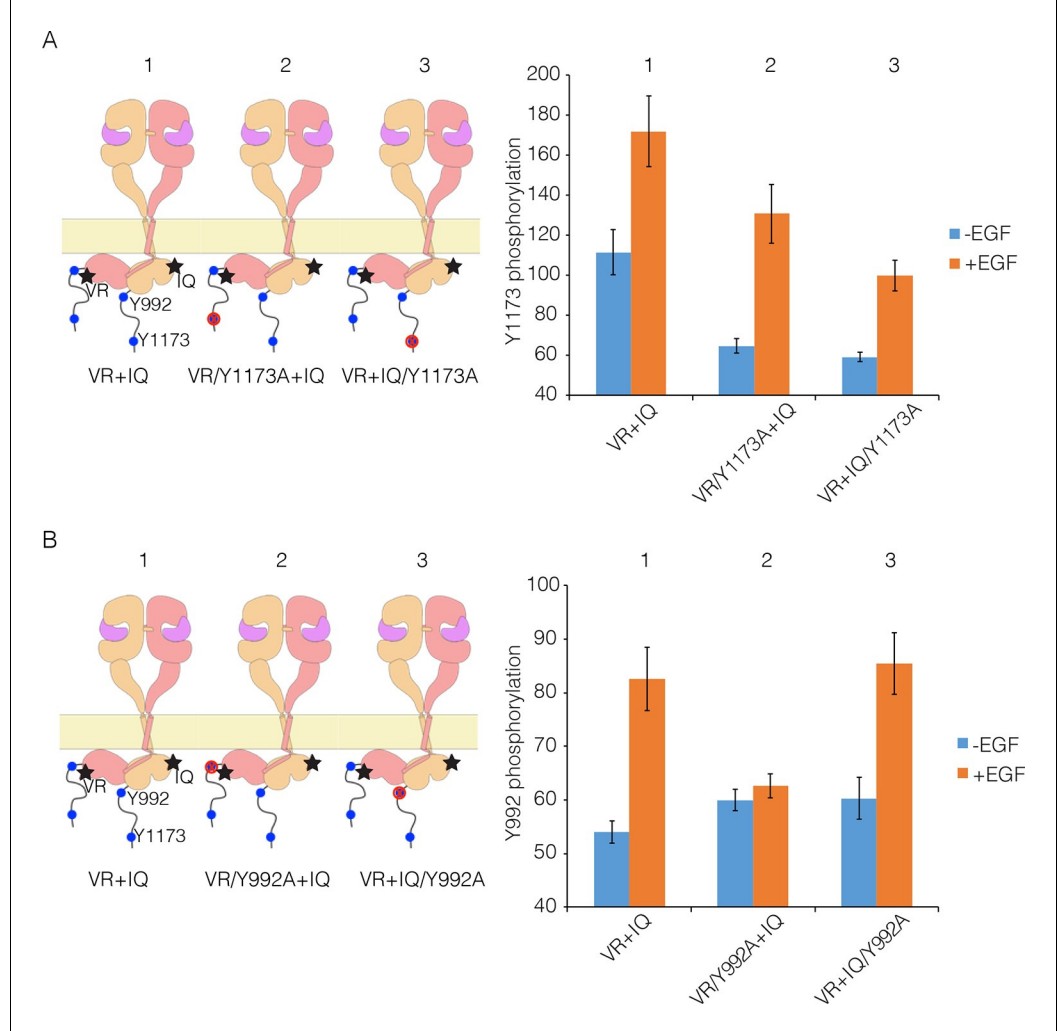

**Figure 9.** Proximal and distal tail phosphorylation of EGFR in co-transfections with activator-impaired and receiver-impaired mutants. In these experiments, activator-impaired EGFR (V924R; denoted VR; salmon subunit) is fused to EGFP at the C-terminus and receiver-impaired EGFR (I682Q; denoted IQ; yellow subunit) is fused to mCherry. Phosphorylation is measured after co-transfection of both constructs, with and without EGF. (**A**) Tyr 1173, located in the distal portion of the tail, is present in both tails (denoted 1), in only the tail of receiver-impaired EGFR (denoted 2) or in only the activator-impaired tail (denoted 3). The bar graphs show phosphorylation levels for Tyr 1173. (**B**) As in Panel A, for a site in the proximal part of the tail, Tyr 992. Note that addition of EGF leads to only a very small increase in phosphorylation of Tyr 992 when the kinase bearing that residue is receiver-impaired.

proximal sites would be phosphorylated efficiently, unless the receiver and activator kinases swapped positions.

In considering the potential relevance of multimerization for EGFR activity, we also wondered about the extent to which activated EGFR molecules ('enzymes') could phosphorylate other EGFR molecules ('bystander substrates') through random encounters, without the necessity for formation of a specific organization. It has been shown previously, using co-transfection of wild-type and chimeric variants of EGFR, that the phosphorylation of bystander EGFR kinase domains does not occur efficiently (**Muthuswamy et al., 1999**). We addressed this issue by co-transfecting active EGFR variants lacking Tyr 1173 in the tail ('Enzyme-EGFR') with EGFR variants that are kinase-dead, but contain Tyr 1173 ('Substrate-EGFR'; see **Figure 10**).

When Enzyme-EGFR is co-transfected with a Substrate-EGFR in which the activator and receiver surfaces are intact, then robust phosphorylation of Tyr 1173 on the Substrate-EGFR is observed (**Figure 10**). Phosphorylation of Tyr 1173 is also observed when the Substrate-EGFR is receiver-impaired

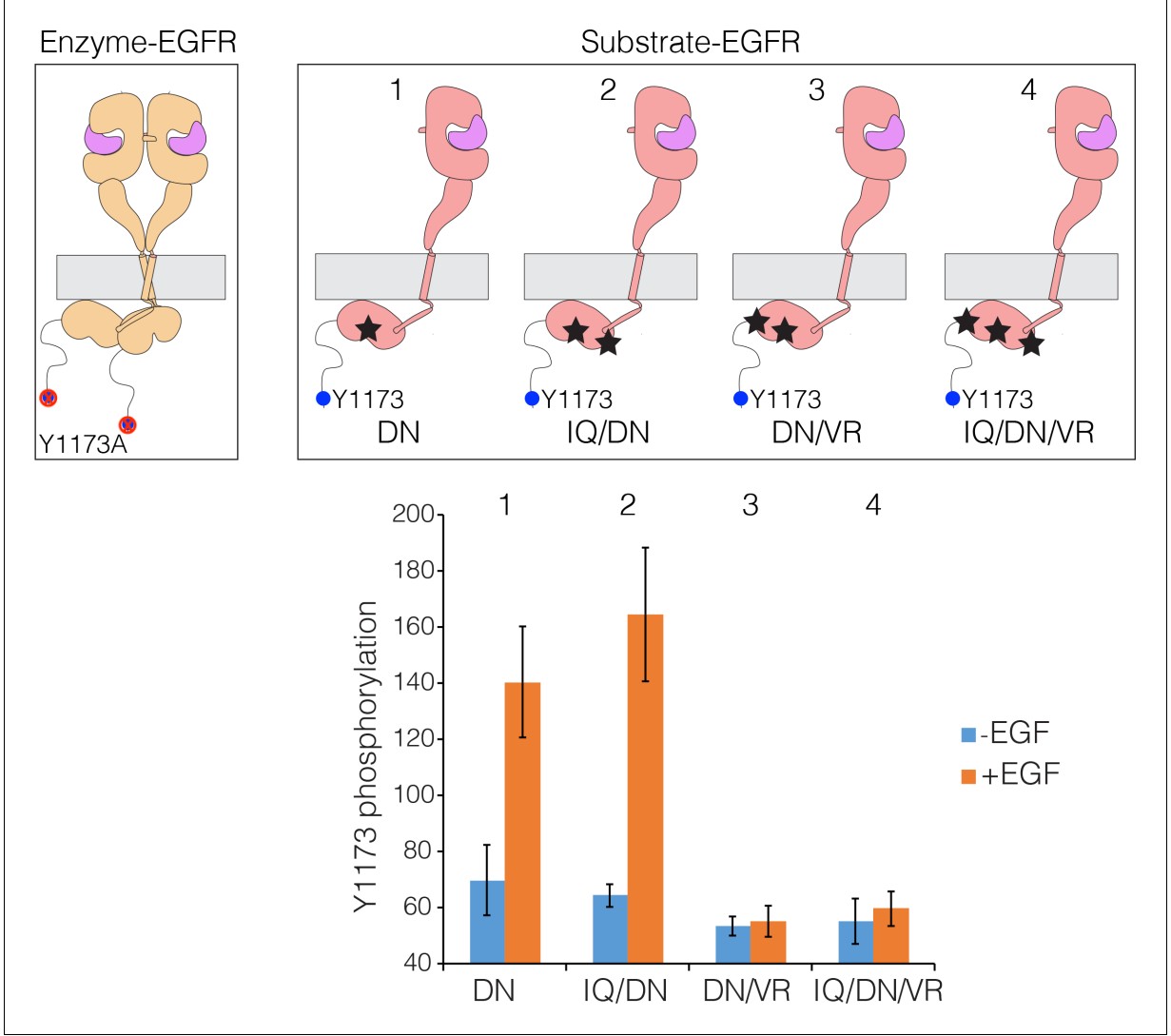

**Figure 10.** Lateral phosphorylation of the tail of EGFR. Top) Schematic of kinase-active EGFR (Enzyme-EGFR) that is co-transfected with a kinase-dead variant (D813N; Substrate-EGFR), as shown in the schematic diagram. Tyr 1173 in Enzyme-EGFR is replaced by alanine. Four variants of substrate-EGFR are used: (1) kinase-dead EGFR, with no other mutations, denoted DN, (2) kinase-dead EGFR that is also receiver-impaired, IQ/DN, (3) kinase-dead EGFR that is also activator-impaired, DN/VR, and (4) kinase-dead EGFR that is also activator- and receiver-impaired, IQ/DN/VR. Bottom) Bar graph showing phosphorylation levels for Tyr1173, with and without EGF. Note that Substrate-EGFR that is activator-impaired shows no increase in phosphorylation above the basal level upon EGF addition.

(I682Q) (*Figure 10*). Strikingly, if the Substrate-EGFR is activator-impaired (V924R), then phosphorylation of Tyr 1173 is not efficient (*Figure 10*). Likewise, phosphorylation of Tyr 1173 is also not efficient if the substrate-EGFR is both activator-impaired and receiver-impaired (*Figure 10*). Although a previous study had inferred that lateral phosphorylation of EGFR tails is possible (*Ruan and Kannan, 2015*), our results lead us to conclude that lateral phosphorylation of the tail in bystander substrates is inefficient, and that for efficient tail phosphorylation to occur, the kinase bearing the tail has to be part of an asymmetric dimer.

These results suggest that multimerization involving chains of asymmetric dimers of kinase domains might be a mechanism for boosting the phosphorylation of both proximal and distal sites on the tail. In such a multimer model, where the majority of kinase domains are sandwiched by other kinase domains, they function as 'activators' and 'receivers' simultaneously. Therefore, phosphorylation of the EGFR tail would be potentiated and more easily propagated in response to EGF.

## A model for EGFR multimers

We have constructed a plausible model for an EGFR multimer that is consistent with the constraints implied by our findings (a summary of the key results that we used in constructing this model is provided in *Figure 11—figure supplement1*). The components of the model are the known dimeric arrangements of the extracellular module, the transmembrane helices and the kinase domains, which are connected together to produce a model for dimeric full-length EGFR, as described (*Arkhipov et al., 2013a*; *Endres et al., 2013*). We then modeled tetramers of EGFR by packing the dimers against each other as rigid bodies. This model is simply a useful construct for considering how tetramerization might occur, and is not based on detailed energy calculations. Also, the limited set of mutations we have made does not allow us to conclude definitively that the residues that we have identified as being important are responsible for a direct and stereospecific interaction between dimers. Thus, the models should be considered as structural hypotheses that await more rigorous experimental tests.

We began by docking two dimers of the extracellular module to generate a tetramer. There are, at present, only two crystal structures for intact extracellular modules of EGFR-family members bound to ligands, EGFR bound to EGF (PDB code: 3NJP; [*Lu et al., 2010*]), and HER4 bound to neuregulin-1β (PDB code: 3U7U; [*Liu et al., 2012*]). The principal difference between these is that the EGFR extracellular module is closed, with the C-terminal tips of the extracellular module are close together, whereas in HER4 they are separated. These differences reflect an intrinsic flexibility in the extracellular module, rather than fundamental difference between EGFR and HER4, as shown by molecular dynamics simulations of EGFR (*Arkhipov et al., 2013a*). We found that a closed structure, with the tips of Domain IV close together, is more suitable for the generation of multimers in which each subunit retains connection to a planar membrane (see *Figure 11—figure supplement 2*).

Using the EGFR dimer (PDB code: 3NJP), we ran ClusPro to generate tetramers, with the constraint that the seven residues in Domain IV that are important for multimerization are at the interface between dimers. This resulted in the generation of a family of tetrameric arrangements of the extracellular module that resemble each other in general (see *Figure 11A* for a representative model for an extracellular tetramer). The ligand-bound heads of the receptors (Domains I to III) are stacked in a parallel arrangement, with a Domain IV leg from one dimer interacting with a Domain IV leg from the other. These dimers are open structures, and higher-order multimers can be formed readily by adding additional dimers through propagation of the Domain IV-Domain IV interaction. We checked that the selected ClusPro solutions did not place any of the documented glycosylation sites at the interface between dimers (*Zhen et al., 2003*).

We used the NMR structure of a dimer of the transmembrane helices and the juxtamembrane-A segment embedded in a lipid bilayer to extend the tetramer model (PDB code: 2M20; [*Endres et al., 2013*]). We connected the N-terminal ends of the transmembrane helices to the C-terminal ends of each Domain IV. Crystal structures of the kinase domains of EGFR have chains in which each kinase domain can serve as the allosteric activator for the next kinase domain, and can be activated by the previous one (*Stamos et al., 2002*; *Zhang et al., 2006*). We generated a chain of four kinase domains based on crystal structures of the EGFR kinase domain (PDB codes: 2GS6 and 3GOP). The first two kinase domains in this model are connected by the juxtamembrane-B latch, which is linked to the dimer of the transmembrane-juxtamembrane-A segments (*Arkhipov et al., 2013a*). The second two kinase domains are connected similarly. The two pairs are joined by the N-lobe/C-lobe interface that defines the asymmetric dimer, without the juxtamembrane latch. The first three kinase domains in the kinase domains are assumed to be active, while the fourth one is not.

A model for an EGFR tetramer in which all of the components are connected as described is shown in *Figure 11B*. All-atom molecular dynamics simulations using this model is extremely demanding of computer time. Instead, we generated a coarse-grained representation of the tetramer in a lipid bilayer using the MARTINI force field (*Marrink et al., 2007*). A 1 microsecond molecular dynamics trajectory using the coarse-grained model indicates that the inter-domain connections and inter-subunit interfaces in the model are plausible, since the tetramer remains intact over this timescale (*Figure 11—figure supplement 3* and *Figure 11—figure supplement 4*).

This model can be extended to build a trimer of EGFR dimers, by repeating the interactions from dimer to dimer, as indicated schematically in *Figure 11C*, but the connection between the kinase domains and the transmembrane helices raises a complication. In structures of active EGFR (e.g.,

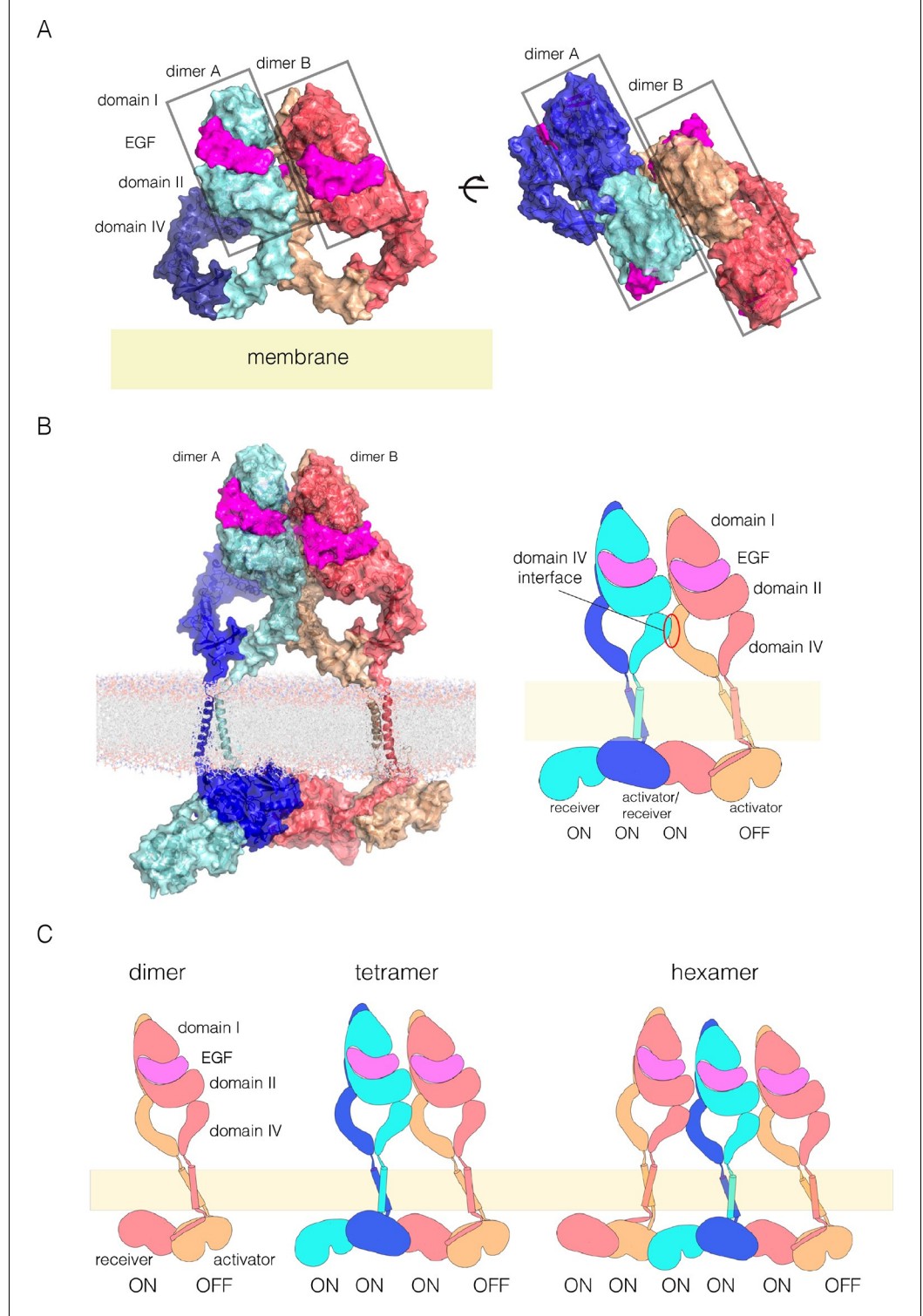

**Figure 11.** A model for EGFR multimerization. (**A**) Two orthogonal views of a tetramer of extracellular modules, selected manually from ClusPro solutions for docking dimers (PDB code: 3NJP) with the constraint that the residues in Domain IV that affect multimerization are located at the interface between dimers. (**B**) A model for an EGFR tetramer, generated by connecting the model shown in Panel A to the structure of the dimeric transmembrane helices (PDB code 2M20) and a chain of kinase domains (PDB codes: 2GS6 and 3GOP). A schematic representation of the tetramer is shown on the right. (**C**) Schematic diagram indicating how higher-order multimers might be generated by repeating the interactions used to generate the model for the tetramer.

*Figure 11 continued on next page*

*Figure 11 continued*

The following figure supplements are available for figure 11:

**Figure supplement 1.** Summary of EGFR variants studied in this work and their effects on EGFR activity and multimerization.

**Figure supplement 2.** The Domain IV interface in crystal structures of the extracellular dimers of EGFR (left; PDB code 3NJP) and HER4 (right; PDB code 3U7U).

**Figure supplement 3.** Coarse-grained molecular dynamics simulation in a lipid bilayer.

**Figure supplement 4.** Structural integrity of the tetramer model in coarse-grained molecular dynamics simulation.

PDB Code 2GS6), the rotation between subsequent kinase domains in a chain is 120°. If this rotation is preserved, then the juxtamembrane dimer has to be broken in the third dimer in order to connect to the transmembrane helices. We note that there are structures of the EGFR, HER2 and HER4 kinase domains in inactive conformations in which chains of asymmetric dimers are formed (*Aertgeerts et al., 2011*; *Qiu et al., 2008*; *Red Brewer et al., 2009*). In these chains, the rotation between adjacent kinase domains is 90°, rather than 120°, due to rotation of the N-lobe of the kinase domain with respect to the C-lobe. This intrinsic flexibility in the kinase domain may allow maintenance of the juxtamembrane interactions along the chain, but we have not investigated this further.

Our model for extended multimers relies on the C-terminal tips of the extracellular module being close together (see *Figure 11—figure supplement 2*). A closed conformation of the extracellular module would be stabilized by the dimerization of the transmembrane helices, which brings the N-terminal ends of the helices close together (*Endres et al., 2013*; *Mineev et al., 2010*). Such coupling may explain the critical importance of the transmembrane dimer for multimerization. In the Xenopus oocyte assay, we found that disruption of the transmembrane dimer interface resulted in a substantial reduction in the extent of multimerization (*Figure 5B*). In contrast, the deletion of the dimerization arm in the extracellular module (Δ-arm) had very little effect. In terms of its effect on activity, EGFR that is mutated in the transmembrane helix behaves similarly to EGFR with Domain IV mutations that block multimerization, with strong inhibition of PI3K phosphorylation, and only a mild effect on ERK phosphorylation (*Figure 8—figure supplement 1*). Deletion of the dimerization arm compromises EGFR function in a more general way, blocking both PI3K and ERK phosphorylation (not shown). This differential effect of mutations in the transmembrane helices on proximal versus distal sites may explain why it has been difficult to observe strong effects on EGFR activity by disrupting transmembrane dimerization (*Endres et al., 2013*; *Lu et al., 2010*).

It is possible that this model for multimerization may apply to other human EGFR family members as well. Examination of the Domain IV multimerization interface shows that although there are changes in the identity of residues within this region, the general hydrophobicity of residues in this surface region is conserved. For heterodimers of HER2 and HER3, an interesting possibility is that a tetramer could have two HER2 kinase domains sandwiched between HER3 kinase domains. The HER2 tail is similar in sequence to that of EGFR, and is likely to be subject to the same constraints on the phosphorylation of proximal sites. The HER3 tail is distinctive, and includes a ~70 residue long spacer between the kinase domain and the first phosphorylation site, making HER3 less dependent on a specific organization of kinase domains for phosphorylation. The first HER2 kinase domain would be activated by HER3, and would serve as the activator for the second HER2 kinase domain. Multimerization has been shown to be important for the phosphorylation of HER2 in HER2-HER3 heterodimers (*Zhang et al., 2012*), and the simultaneous activation of two HER2 kinase domains in a tetramer might account for this.

## Concluding remarks

Our studies on the stoichiometry of EGFR show that the receptor is predominantly monomeric in the absence of EGF, at low expression levels. More importantly, we found that ligand-induced receptor self-association does not end at dimerization, but continues with the generation of multimers. While

the existence of higher-order ligand-induced multimers of EGFR has been reported before (*Clayton et al., 2005*; *2008*), our identification of residues in Domain IV that specifically affect multimerization rather than dimerization has allowed us to uncover a functional role of multimerization in EGFR activation. The long C-terminal tail of EGFR has two kinds of phosphorylation sites. Those that are distal to the kinase domain (e.g., Tyr 1068, and Tyr 1173, which recruit the adaptor proteins GRB2 and SHC) are phosphorylated more efficiently than those in the proximal part of the tail (*Koland, 2014*; *Kovacs et al., 2015b*). We now show that multimerization is necessary for robust phosphorylation of PI3K, which is recruited to the proximal part of the tail.

We propose a model for an EGFR tetramer that is consistent with our experimental observations. This model is reminiscent of the arrays formed by the sensor of the unfolded protein response, IRE1, in which oligomerization of the peptide-binding module in the endoplasmic reticulum is coupled to the chaining of kinase domains in the cytoplasm (*Korennykh et al., 2009*). In IRE1, the chaining of kinase domains occurs in a back-to-front manner, whereas in EGFR our model has the kinase domains chained in a head to tail fashion, generating a continuous array of activator and receiver kinase domains. Such an array might boost tail phosphorylation by stabilizing the active conformations of the majority of kinase domains, and also by facilitating the phosphorylation of the proximal segments of the tail *in cis*.

Our understanding of the mechanism for EGFR activation relies heavily on structural information for fragments of the receptor. However, the efficient activation of EGFR and the initiation of signaling pathways on the cell membrane involves dynamic interactions between intact receptors and their surroundings, including membranes, adaptor proteins, effector proteins, and the cytoskeleton. The multimerization of EGFR may be crucial for signaling in this environment, since it increases the effective density of docking sites for downstream adaptor proteins. These adaptors are typically multivalent, and their docking could synergize with EGFR multimerization, leading to larger-scale clusters. The formation of such signaling platforms may be a central aspect of maintaining the balance between ligand-induced activation and inactivation by receptor internalization, phosphatases or ubiquitin ligases, and will continue to serve as a focus of intense study.

## Materials and methods

### Single-molecule imaging in live Xenopus oocytes

For experiments in Xenopus oocytes, the human EGFR gene was cloned into pGEMHE-X-EGFP vector at SalI and SacII site to generate a C-terminally EGFP tagged version of EGFR. Messenger RNA (mRNA) was then synthesized in vitro from linearized DNA using mMessage mMachine kit (Ambion). For single molecule imaging by TIRF microscopy, mRNA injection for EGFR-EGFP was optimized for expression. We found that 0.25–0.5 ng of mRNA for approximately 20 hr of incubation at 18°C produced uniform diffraction-limited spots when imaged by TIRF microscopy.

In the absence of EGF, imaging of EGFR-EGFP on Xenopus oocyte membrane by TIRF microscopy was performed as previously described (*Chen et al., 2015*; *Ulbrich and Isacoff, 2007*). Briefly, 10–20 oocytes were manually devitellinized after ~20 hr of expression at 18°C, placed on high refractive index coverslips (n = 1.78; Olympus, Japan) and excited using a phoxX 488 (60 mW) laser and, imaged using an Olympus 100×, numerical aperture (NA) 1.65 oil immersion objective at room temperature. A 495 nm long-pass dichroic mirror was used at excitation in combination with a 525/50 nm band-pass filter at emission. Five hundred to eight hundred frames at the rate of 20 Hz were acquired using an electron multiplying charge coupled device (EMCCD) camera (Andor iXon DV887).

The Xenopus oocytes flatten on the surface of coverslips, and we found that if we added fluorescently labeled EGF to oocytes that had already been placed on the coverslips there was limited access of EGF to the receptors. To overcome this problem we added EGF directly into the imaging buffer first, and then placed the oocytes onto the coverslip of the imaging chamber that contains EGF buffer. The EGF that is used in all other experiments is unlabeled. We noticed that upon addition of EGF the spots corresponding to the receptor were no longer fixed in space but were mobile, which complicates the analysis of photobleaching. We found that the addition of a FLAG-tag to the EGFR at the N-terminus resulted in minimization of motion of the receptor, presumably due to better adherence to the glass. We verified that the inclusion of the FLAG tag did not affect the activity

of the receptor as assessed by autophosphorylation in response to EGF in mammalian cells (COS-7) (data not shown).

To observe the effect of EGF on EGFR stoichiometry, we placed the oocytes expressing EGFR-EGFP one by one directly in the chamber containing 15 nM EGF solution in ND96 buffer (5 mM HEPES/NaOH, 96 mM NaCl, 2 mM KCl, 1 mM MgCl2, 1.8 mM CaCl2, pH 7.6) placed on the TIRF microscope. Imaging was started almost immediately, with a maximum delay of 1–2 min, using the same protocol described above.

Only single, immobile, and diffraction-limited spots were analyzed. The number of bleaching steps was determined manually for each single spot included in the analysis; 1000–2000 spots from 5 to 10 oocytes from three to five different batches were analyzed for most of the constructs.

## Pulsed interleaved excitation fluorescence cross-correlation spectroscopy (PIE-FCCS)

Fluorescence correlation spectroscopy measurements were done on a modified inverted microscope (Nikon Eclipse Ti; Nikon Instruments) with customized illumination and detection paths (*Comar et al., 2014*). The sample was illuminated with two overlapping laser beams picked from a 5 ps pulsed supercontinuum fiber source (SuperK NKT Photonics, Birkerod, Denmark). The two beams were spatially separated from the parent white light source with a series of dichroic mirrors and respectively filtered through a 488 nm (LL01-488-12.5, Semrock, Rochester, NY) and a 561 nm (LL02-561-12.5, Semrock, Rochester, NY) cleanup filter. The 561 nm beam was directed through an 18 m single mode fiber, and the 488 nm beam through a 3 m fiber, which introduced a 50 ns time delay for pulsed interleaved excitation. After exiting the fibers, the two beams were spatially overlapped and sent to a dichroic filter block (zt488/561rpc and zet488/561m, Chroma Technology Corp. Bellows Falls, VT) housed within the filter wheel of the microscope. The epifluorescence signal collected by the objective passed through a 50 µm pinhole positioned outside a camera port of the microscope and was split onto two detectors filtered with a 612/69 nm filter (FF01-621/69–25, Semrock, Rochester, NY) and a 520/44 nm filter (FF01-520/44–25, Semrock, Rochester, NY). The signal was detected with single photon avalanche diodes (SPAD) with a 50 µm active area, 30 ps timing resolution, and 25 dark counts per second (Micro Photon Devices, Bolzano, Italy); single photon counts were recorded with a four-channel-routed time-correlated single photon counting (TCSPC) device (Picoharp 300, PicoQuant, Berlin, Germany).

Measurements were made on live Cos-7 cells held at 37°C with an on-stage incubator. The laser focus was positioned near the cell periphery to avoid signal from immobile or slow moving vesicles and organelles. For each cell, data was collected in five successive 15 s increments. The time-tagged data in the each of the 15 s data files is filtered according to the photon arrival time. Photons arriving at the 612/69 nm filtered detector within 50 ns of a 488 nm laser pulse are rejected from the 'red' fluorescence intensity trace, and photons arriving at the 520/44 nm filtered detector within 50 ns of a 561 nm laser pulse are rejected from the 'green' fluorescence intensity trace. This generates a time-dependent fluorescence signal, free from green-to-red bleed-through, FRET, and direct mCherry excitation by the 482 nm laser. Fluorescence correlation and cross-correlation spectra are calculated as described previously. (*Comar et al., 2014*)

## Analysis of phosphorylation by flow cytometry

Human EGFR (UniProt accession no. P00533) constructs were cloned into vectors based on the pEGFP-N1 plasmid (Clontech, Mountain View, CA) using XhoI and SacII restriction sites. In co-transfection experiments, EGFP was replaced by monomeric variants of mCherry (Clontech) using BamHI and NotI sites. Cos-7 or HEK-293T cells grown on 6-well plates were transfected with FuGENE (Promega) and serum starved for 12 hr. These two cell lines were chosen because of their low levels of expression of endogenous EGFR and high transfection efficiency. Cells were dissociated by enzyme-free cell dissociation buffer (Gibco, Thermo Fisher Scientific, Waltham, MA), transferred to 96-well plates, and spun down at 1800 rpm for 3 min. Samples were treated with 100 ng/ml EGF (Sigma-Aldrich, St. Louis, MO) for 2 min at room temperature. After stimulation, cells were immediately fixed in 2% formaldehyde for 10 min. Samples were spun down and permeabilized in ice-cold methanol for 30 min on ice. After rehydration in staining buffer (phosphate-buffered saline (PBS), supplemented with 0.2% bovine serum albumin (BSA) and 1 mM EDTA), cells were stained with

1:100 dilution of primary antibody. Anti-EGFR-pY992 (antibody to phosphorylated EGFR [phosphorylated tyrosine at position 992]) (catalog no. 44-786G) was purchased from ThermoFisher Scientific. Anti-PI3Kinase-p85 (pTyr458)/p55 (Tyr199) (catalog no. 4228) and anti-pErk1/2-pT202/pY204 (catalog no. 4370) antibodies were purchased from Cell Signaling Technology. Anti-EGFR pY1173 (catalog no. sc-12351) was purchased from Santa Cruz Biotechnology. Samples were then stained with 1:100 dilution of anti-rabbit Alexa647-labeled secondary antibody (ThermoFisher Scientific, catalog no. A-21245).

Flow cytometry analysis was done using a BD Bioscience LSR Fortessa cell analyzer (San Jose, CA). EGFP was excited by a 50mW, 488 nm, Coherent Sapphire laser, and emission was detected using a 525/50 nm band-pass filter. A 50mW, 561 nm, Coherent Sapphire laser was used to excite mCherry, and fluorescence was detected using a 610/20 nm band-pass filter. Alexa647 was excited by a 50 mW, 640 nm laser.

In experiments where only one type of receptor was transfected, cells were binned based on their EGFR-EGFP expression levels, and the mean and standard error values were calculated for the antibody-labeling channel (Alexa647) to obtain a receptor expression dependent phosphorylation plots. In co-transfection experiments, cells were gated first by their expression levels of the two cotransfected constructs, one labeled with mCherry and the other with EGFP. Only cells that are simultaneously expressing comparable levels of mCherry and EGFP, which are majority of the cells, are binned based on their expression levels of the EGFR-EGFP. Mean and standard error values for the Alexa647 fluorescence intensities, corresponding to phosphorylation levels, were calculated in each bin.

## Analysis of EGFR surface expression by fluorescence microscopy

Human EGFR was modified with an N-terminal 3 x HA tag (after the signal sequence) and a C-terminal Venus fluorescent protein. HEK-293T cells were transfected with this construct by CaPO4 and plated at 75% confluency on Poly-D-lysine coated 1.5 mm glass bottom 10 cm dishes. 48 hrs after transfection, the cells were starved in staining media (clear MEM supplemented with 10 mM HEPES pH 7.4, 2 mM L-glutamine and 5% BSA) for 1 hr then stained on ice for 45 min with rabbit anti-HA (Cell Signaling, catalog no. 3724) at a 1:1000 dilution in staining media. Cells were then fixed in 4% paraformaldehyde, blocked for 30 min in staining media and secondary antibody staining (Alex-fluor 647 conjugated goat anti-rabbit, ThermoFisher, A-21244) at 1:1000 dilution in staining media. High resolution images were collected on a Nikon TE-2000 inverted microscope with a 100X objective. Venus was pseudocolored cyan and 647 was pseudocolored magenta. Images for quantification were collected with a 20X objective and analyzed automatically using Nikon's NIS-elements software to determine regions of interest (ROI) based on intensity thresholds of the total (Venus) receptor channel. Surface/total ratio of each ROI was averaged for each day and plotted for one representative experiment. The same camera and laser settings were used for all images collected.

## Docking of EGFR by ClusPro

The crystal structure of the dimer of extracellular domain of EGFR bound to EGF (PDBID 3NJP) was docked against itself using ClusPro (*Kozakov et al., 2013*). For initial docking, we excluded the Domain IV 'leg' so that docking was carried out using just the Domains I, II and III 'head' regions. We then used Cluspro to dock the entire extracellular domain against itself, with the added constraint that the residues identified by mutation studies to play a role in oligomerization (Val 526, Glu 527, Asn 528, Ile 545, Thr 548, Asn 554 and Ile 556) are at the interface. We manually filtered the ClusPro solutions using three criteria: (i) the C-terminal ends of both dimers must point towards to same plane, (ii) the rotation between the dimers must be small enough to permit the multimer to be extended while retaining connection to the same plane and (iii) documented sites of glycosylation (*Zhen et al., 2003*) should not be buried at the interface. Application of these filters yielded a family of 6 models that differed from each other only with respect to the tilt and rotation between two dimeric units. We chose a representative one to construct a model for a tetramer of full-length EGFR.

## Coarse-grained simulation of EGFR tetramer

A representative model of the tetramer was constructed as described in the main text. Missing residues in the connections between different domains were built using PyMOL (The PyMOL Molecular Graphics System, Version 1.8 Schrödinger, LLC.). The complete tetramer structure was energy-minimized using an all-atom force field in the program AMBER (*Case et al., 2005*). The model was converted to a coarse-grained representation using the *Martinize* program (*Monticelli et al., 2008*) and inserted into a lipid bilayer comprised of Dipalmitoylphosphatidylcholine (DPPC) and solvated using the I*nsane* program (*Wassenaar et al., 2015*). The final system comprised 1491 and 1461 DPPC molecules in the lipids in the extracellular-facing and intracellular-facing leaflets of the membrane, respectively. The system had $Na^+$ and $Cl^-$ ions at 0.15 M.

GROMACS (*Pronk et al., 2013*) was used to generate a molecular dynamics trajectory using the coarse-grained representation and the MARTINI force field (*Monticelli et al., 2008*). The overall structures of the individual EGFR subunits and EGF was maintained with an elastic network model, with no such constraints applied between different subunits. Harmonic restraints were applied between pairs backbone atoms within a subunit, with a bond cut-off between 5 Å and 9 Å, and a force constant of 5 kJ $mol^{-1}Å^{-2}$. The energy of the system was minimized using the steepest descent method, first while holding the protein fixed, and then while allowing all the atoms to move. The lipid bilayer was then allowed to equilibrate while holding the protein fixed for 40 ns. Pressure was held constant at 1 bar using the Berendsen algorithm (*Berendsen et al., 1984*) with a compressibility of 5.0 x $10^{-5}$ $bar^{-1}$ and a time constant of 10 ps. Pressure control was applied independently in the plane of the membrane, and normal to it. Temperature was maintained at 323 K by rescaling the velocity of all particles (*Bussi et al., 2007*) with a time constant of 1 ps. Electrostatic interactions were shifted to zero between 0.0 Å and 12.0 Å. Van der Waals interactions were shifted to zero between 9.0 Å and 12.0 Å. A time step of 20 fs was used.

The system was then allowed to run for 1 μs in at constant temperature (T=323 K) and pressure (P=1 bar) using the same parameters as for the equilibration run, with the following differences. Pressure was controlled using Parrinello-Rahman algorithm (*Parrinello, 1981*). Electrostatic interactions were shifted to zero between 0.0 Å and 11.0 Å. Van der Waals interactions were shifted to zero between 9.0 Å and 11.0 Å.

## Acknowledgement

We thank members of the Kuriyan lab and Isacoff lab for helpful discussions and comments on the manuscript. We thank Hector Nolla and Alma Valeros of Flow Cytometry Facility at UC Berkeley, for technical help with the FACS instruments. We thank Zhu Fu of the Isacoff lab for technical help with the cloning. We thank Kevin Skinner and Morgan Torcasio of the Smith lab for technical help with cell culture and the PIE-FCCS experiments. We thank David E Shaw and Yibing Shan for insightful discussions on EGFR mechanism, and Yamuna Krishnan for critical comments on the manuscript. YH is supported by the Howard Hughes Medical Institute International Student Research Fellowships Program.

## Additional information

### Competing interests

JK: Senior editor, *eLife*. The other authors declare that no competing interests exist.

### Funding

| Funder | Grant reference number | Author |
|---|---|---|
| National Cancer Institute | 2RO1CA09650406 | John Kuriyan |
| National Institutes of Health | R15EY024451 | Adam W Smith |
| National Institutes of Health | 1R01GM117051 | Ehud Y Isacoff |

The funders had no role in study design, data collection and interpretation, or the decision to submit the work for publication.

## Author contributions

YH, Designed Xenopus oocyte based single-molecule analysis of EGFR stoichiometry and interpreted the data, Conducted the data collection and analysis, Designed mammalian cell-based EGFR activity assay and conducted the data analysis and interpretation, Designed and conducted docking and model building, Drafting or revising the article, Contributed unpublished essential data or reagents; SB, Designed Xenopus oocyte based single-molecule analysis of EGFR stoichiometry and interpreted the data, Conducted the data collection and analysis, Drafting or revising the article; DK, Designed and conducted docking and model building, Acquisition of data, Analysis and interpretation of data, Drafting or revising the article; SMP, Designed mammalian cell-based EGFR activity assay and conducted the data analysis and interpretation, Acquisition of data, Drafting or revising the article; MM, XS, MJK, Designed mammalian cell-based PIE-FCCS experiments and conducted the data collection, analysis and interpretation; AWS, Designed mammalian cell-based PIE-FCCS experiments and conducted the data collection, analysis and interpretation, Drafting or revising the article; EYI, Designed Xenopus oocyte based single-molecule analysis of EGFR stoichiometry and interpreted the data, Drafting or revising the article; JK, Designed Xenopus oocyte based single-molecule analysis of EGFR stoichiometry and interpreted the data, Designed mammalian cell-based EGFR activity assay and conducted the data analysis and interpretation, Designed and conducted docking and model building, Drafting or revising the article.

## Author ORCIDs

Ehud Y Isacoff, http://orcid.org/0000-0003-4775-9359
John Kuriyan, http://orcid.org/0000-0002-4414-5477

## Ethics

Animal experimentation: This study was performed in strict accordance with the recommendations in the Guide for the Care and Use of Laboratory Animals of the National Institutes of Health. Our Xenopus protocol has been approved by Institutional Animal Care and Use Committee (IACUC) of the University of California.

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
