## [Decision Letter]

Thank you for submitting your work entitled "Molecular basis for multimerization in the activation of the Epidermal Growth Factor Receptor" for consideration by *eLife*. Your article has been reviewed by three peer reviewers, and the evaluation has been overseen by a Reviewing Editor (Volker Dötsch) and Ivan Dikic as the Senior Editor.

The reviewers have discussed the reviews with one another and the Reviewing Editor has drafted this decision to help you prepare a revised submission.

Summary:

The study of Huang et al. addresses the nature and importance of higher order EGFR complexes. The existence of these higher order complexes have been described previously but their function, existence, and significance have been questioned and ill defined. Part of the conundrum was due to our increased understanding of EGFR structure and function that has been gained since 2002. The field now has a detailed appreciation of extracellular and intracellular interactions that contribute to EGFR activation. However, the observation of higher order complexes complicated the interpretation of this elegant structural understanding. The present study enlightens the field as to the structural basis for these higher order complexes. In summary, this manuscript reports on some very important findings and will contribute to our basic understanding of EGFR signaling.

However, during the review several questions were raised that require revision of the manuscript.

Essential revisions:

1) The description of the modelling seems too long and should be shortened or moved in part to supplementary material. In addition, the models in Figure 11 and Figure 12 are a concern. Although the authors stress that it is not based on detailed energy calculations, the model is too precise in what it states. Yet, given that it is not based on detailed energy calculations, it must certainly be inaccurate. It would be better to remove all but the bottom right cartoon in Figure 11.

2) In experiments depicted in Figure 3, the authors subtract 3% as the baseline multimerization rate based on experiments with Claudin. It is not clear how this subtraction is done, but if there is no consideration of the error in the 3% value then the error in the value after subtraction is underestimated. Germane to that point the change in 2-step percentage from 3% to 1.5% when the V924R mutation is introduced (Figure 3) appears well within the (perhaps too small) error and not persuasively significant. The conclusion that the asymmetric kinase dimer contributes to mediation of the "preformed" EGFR dimers thus does not seem justified. The effects of the V924R mutation on the increase in preformed dimers caused by Δ-arm, Δ-tether, and L834R mutations appears more robust, but it is a bit puzzling that V924R should have an effect on the Δ-arm and Δ-tethering as such mutations do not appear to result in higher activity of the receptor, which suggests the deletions results in an inactive dimer (Ogiso/Garrett/Schlessinger).

Similarly, is the difference in multistep events seen for TGFa (in Figure 4) and EGF (in Figure 4) statistically significant? It seems to be as strong as some of the other changes reported.

3) The data in Figure 6 lacks crucial controls. Only the IIIV/AARE has no influence, apparently suggesting that I545 and I556 can accommodate alanines. To strongly argue that the data in Figure 6 identify a 'patch' involved in multimerization, mutations outside the patch that have no influence should be looked at.

4) The method used to assess EGFR autophosphorylation in Figure 8 does not allow assessment of whether cell surface expression of the mutants is a problem. Has this been looked at? If the mutants are impaired in cell surface expression, they will not be activated by EGF.

5) The data in Figure 8 do not seem to be a compelling argument that 'multimerization affects proximal phosphorylation of the EGFR tail more strongly than distal phosphorylation'. To generalize like this, more sites need to be looked at. Other influences on multimerization should be assessed for the same effect too. For example, is there any reason that the region identified on domain IV could not simply be important for mediating clustering prior to endocytosis-and not be responsible for a stereospecific multimer?

Detection of phosphorylation is also based on assessing receptor phosphorylation in a transient expression system. Mass spectrometry data would be a more direct and convincing method to show the effect. Could stably expressing cell lines instead of transient expression be used?

Similarly, the arguments surrounding Figure 10 have to be clarified. What these data show is simply that loss of 'activator' function leads to loss of Y1173 phosphorylation. How does this translate into an argument that multimerization might be a mechanism for boosting phosphorylation of proximal and distal sites on the tail?

6) Likewise does dimerization arm deletion increases two-step bleaching events in any specific way, or is this a sign of aggregation. Does this form still bind EGF the same?

7) The specificity of the P-Tyr specific antibodies should be demonstrated using EGFR mutants Y955F, Y992F, Y1173F, Y1186F etc.

8) Could the GFP fusion partner alter the association/dissociation of EGFR? This is especially important as it pertains to the formation of the higher order complexes. Indeed the authors point out that their previous work with GCN4 fusions led to some unexpected associations. Could an experiment or two with antibody (Fab) Q-dots with non-GFP fused receptors be helpful?

---

## [Author Response]

In considering all of the points raised by the reviewers, we were struck by the importance of one issue that had not been addressed in our submitted manuscript, but was raised by the review. This issue concerned whether the mutations we had made had affected the surface display and trafficking of the receptor (Point No. 4 in the list of essential points, below). As the reviewers point out, if the mutations affect the surface expression of EGFR, then the reduction in EGF- induced activity would be difficult to interpret. We are pleased to report that we carried out new experiments that addressed this issue for one of the key mutations (denoted II/KK) for which we show a reduction in activity, and we demonstrate that there is no effect on surface expression of EGFR. Briefly, we made a construct of EGFR which has an HA tag at the N-terminus of the extracellular domain, and has the fluorescent protein Venus fused to the C-terminal end. We detected surface expression by addition of an HA-specific antibody to intact live cells. We then fixed the cells and examined them by fluorescence microscopy, and quantitated the ratio of surface-expressed EGFR to total EGFR (monitored by Venus fluorescence). There is no discernible difference in this ratio between wild-type EGFR and mutant EGFR. These data are now included in the discussion of the effects of Domain IV mutations on EGFR activity (Figure 8—figure supplement 2). Sean Peterson, who carried out these new experiments, is now included as a co-author of the revised manuscript.

Another new set of data and analysis concerns the pulsed-interleaved fluorescence cross- correlation spectroscopy (PIE-FCCS) measurements has been added to the revised manuscript. Additional PIE-FCCS data on the constructs reported in our original submission have been measured, and these data are shown as Figure 7 in the revised manuscript. In addition, diffusion constants have now been extracted from PIE-FCCS data, and are included as a new Figure 7. The mobility of each construct in the plasma membrane is directly measured with PIE-FCCS and is converted to an effective diffusion coefficient, D_eff_, for comparison. The diffusion coefficients of EGFR and EGFR-IIIV/KKRE before ligand stimulation are 0.60 ± 0.02 and 0.59 ± 0.02 μm^2^/s respectively ( ± standard error). These values are consistent with monomeric, single-pass transmembrane proteins in the plasma membrane (Marita et al., Biophys J, 2015, PMID: 26536270). Upon stimulation with EGF, the diffusion coefficient of EGFR-IIIV/KKRE drops to 0.52 ± 0.02 μm^[2]^/s, consistent with a monomer-dimer transition. For wild-type EGFR, EGF stimulation causes the diffusion coefficient to drop to 0.36 ± 0.01 μm^[2]^/s. This drop is significantly more than seen in the mutant and consistent with the formation of multimers. Finally, the PIE- FCCS data have been displayed in an alternative way to help the reader understand these data better (Figure 7—figure supplement 1, also shown below).

A detailed response to each of the comments of the Reviewers is given below. Text extracted from the review is shown in blue, and our responses are in black. We also provide a document in which changes made to the original manuscript are highlighted.

*Essential revisions:*

*1) The description of the modelling seems too long and should be shortened or moved in part to supplementary material. In addition, the models in Figure 11 and Figure 12 are a concern. Although the authors stress that it is not based on detailed energy calculations, the model is too precise in what it states. Yet, given that it is not based on detailed energy calculations, it must certainly be inaccurate. It would be better to remove all but the bottom right cartoon in Figure 11.*

In response to this comment we have moved Figure 12A (results of coarse-grained molecular dynamics simulations) to supplemental material (Figure 11—figure supplement 3).

We feel that it is important to retain one illustration of a molecular model for the multimers in the main figures, otherwise we worry that our arguments will be difficult to follow. We have therefore retained Figure 11 in the main figures. Note that these are “collages” of actual crystal structures (or NMR structures) of the components that are connected together in a hypothetical way. The detail arises from the high resolution of the underlying component structures. The cartoon for multimerization that was originally Figure 12B is now Figure 11. The original Figure 12—figure supplement 1 has been moved to Figure 11—figure supplement 4.

While we agree that the modeling we have done is “coarse”, and not much removed from packing physical models together to determine shape complementarity, we feel that it is better to retain the full description in the main body of the paper, so that readers can better assess what we have done. We worry that if we move this to Supplemental Material then the nuances of the modeling risk getting lost.

*2) In experiments depicted in Figure 3, the authors subtract 3% as the baseline multimerization rate based on experiments with Claudin. It is not clear how this subtraction is done, but if there is no consideration of the error in the 3% value then the error in the value after subtraction is underestimated. Germane to that point the change in 2-step percentage from 3% to 1.5% when the V924R mutation is introduced (Figure 3) appears well within the (perhaps too small) error and not persuasively significant. The conclusion that the asymmetric kinase dimer contributes to mediation of the "preformed" EGFR dimers thus does not seem justified. The effects of the V924R mutation on the increase in preformed dimers caused by Δ-arm, Δ-tether, and L834R mutations appears more robust, but it is a bit puzzling that V924R should have an effect on the Δ-arm and Δ-tethering as such mutations do not appear to result in higher activity of the receptor, which suggests the deletions results in an inactive dimer (Ogiso/Garrett/Schlessinger).*

*Similarly, is the difference in multistep events seen for TGFa (in Figure 4) and EGF (in Figure 4) statistically significant? It seems to be as strong as some of the other changes reported.*

We did make the appropriate adjustments to the estimated errors in Figure 3. Nevertheless, we agree with the reviewers that the effect of the V924R mutation on wild-type and mutant EGFR (without EGF) is unexpected. This is an interesting issue, but is not really relevant to the multimerization of EGFR in the presence of EGF. We have therefore deleted the entire discussion of the effects of the V924R mutation on EGFR without EGF. Panels D, E, F and G of Figure 3 have been removed.

*3) The data in Figure 6 lacks crucial controls. Only the IIIV/AARE has no influence, apparently suggesting that I545 and I556 can accommodate alanines. To strongly argue that the data in Figure 6 identify a 'patch' involved in multimerization, mutations outside the patch that have no influence should be looked at.*

We agree that we have not really restricted the potential interaction region to a “patch” that is clearly defined, because we have analyzed relatively few mutations. We have changed the language in the paper as follows:

The section heading has been changed to: “Identification of residues in Domain IV that are necessary for EGFR multimerization”, from “Identification of a region […]”. We have removed the use of the word “patch” in this section, focusing instead on the identification of residues that have an effect on multimerization. We end this section with a new sentence: “We note that we have tested only a limited set of mutations, and that therefore our list of residues within a putative interfacial region is likely to be incomplete.”

In terms of a “control”, we point out, as the reviewer recognizes, that the IIIV/AARE mutation, in Domain IV, does not reduce multimerization substantially. Likewise, mutation of the dimerization arm also behaves similarly. Thus, not all mutations in the extracellular module decrease multimerization.

We agree that a much more exhaustive set of mutations is required in order to gain confidence that we have indeed defined the full extent of the interfacial patch. The experimental tests of multimerization that we employ are very time consuming. Since receiving the reviews we have analyzed two additional mutations using the *Xenopus* oocyte single molecule assay. The first is a single point mutation (Lys 454 to Glu) in Domain III, at a location predicted to be at the interface between the ligand-bound heads of extracellular dimers. This results in a reduction in multimerization (from ~46% in wild type to ~26% in the mutant; see Figure below). We also tested a mutation at the C-terminal tip of Domain IV, in a region close to where the dimerization arm is docked in the autoinhibited conformation of the ligand-free extracellular domain. This mutation (Leu 595 to Arg) also results in a reduction in multimerization (to ~16%). This reduction cannot be accounted for by the dimer-dimer interfaces in our model. While we do not understand the origin of this effect, we note that the C-terminal tip of Domain IV is close to the transmembrane helices, which we have shown to be important for multimerization.

Our feeling is that without a large number of such mutations we cannot substantially advance our understanding of the multimers, so we prefer to hold these results for a later study rather than include them in the revised manuscript.

*4) The method used to assess EGFR autophosphorylation in Figure 8 does not allow assessment of whether cell surface expression of the mutants is a problem. Has this been looked at? If the mutants are impaired in cell surface expression, they will not be activated by EGF.*

This is an extremely important point. We have now demonstrated conclusively that there is no difference in surface expression of the II/KK mutant EGFR, which shows reduced activity, as described in the opening paragraph above. Please see Figure 8—figure supplement 2, and the discussion in paragraph three, subsection “Functional role of multimerization in EGFR autophosphorylation and downstream signaling” of the revised manuscript. We thank the reviewers for pointing out this important issue.

*5) The data in Figure 8 do not seem to be a compelling argument that 'multimerization affects proximal phosphorylation of the EGFR tail more strongly than distal phosphorylation'. To generalize like this, more sites need to be looked at.*

We agree that it is too strong to say that proximal sites are affected more strongly than distal sites. Throughout the manuscript, we have taken care to remove such statements. For example, in the revised manuscript, we now say: “These mutations reduce autophosphorylation of the C-terminal tail of EGFR and attenuate phosphorylation of phosphatidyl inositol 3-kinase, which is recruited by EGFR.”, where previously we said: “These mutations reduce EGFR autophosphorylation in mammalian cells, particularly for sites in the C-terminal tail that are proximal to the kinase domain, and they also attenuate phosphorylation of phosphatidyl inositol 3-kinase, which binds to a proximal site.”

We now say: “Each of three mutants show reduced phosphorylation of both sites compared to wild-type.”, where previously we said: “Each of three mutants show reduced phosphorylation of both sites compared to wild-type, with the effect being more pronounced for the proximal site (Tyr 992).”

*Other influences on multimerization should be assessed for the same effect too. For example, is there any reason that the region identified on domain IV could not simply be important for mediating clustering prior to endocytosis-and not be responsible for a stereospecific multimer?*

We do not believe that endocytosis-related clustering is a component of the multimerization. We observe no multimer in the absence of ligand, so the continual internalization and recycling that the receptor undergoes in the absence of EGF is not responsible for the multimerization. When we add EGF to a kinase dead EGFR (Figure 4), we still observe multimers. So the multimerization does not require an active response to endocytosis machinery (e.g., through phosphorylation). Thus, we believe that the multimerization is a direct structural response of the receptor, without involving downstream factors. We do agree that we have not demonstrated that the multimerization is stereospecific. We now state that this is the case in the revised manuscript. At the beginning of the section on modeling we now say:

“Also, the limited set of mutations we have made does not allow us to conclude definitively that the residues that we have identified as being important are responsible for a direct and stereospecific interaction between dimers. Thus, the models should be considered as structural hypotheses that await rigorous experimental tests.”

*Detection of phosphorylation is also based on assessing receptor phosphorylation in a transient expression system. Mass spectrometry data would be a more direct and convincing method to show the effect. Could stably expressing cell lines instead of transient expression be used?*

The activity of EGFR is dependent on the surface density of EGFR. As we have shown in detail in two previous publications (Endres et al., Cell, 2013, PMID: 23374349 and Kovacs et al., MCB, 2015, PMID: 26124280), by measuring both the expression level and the activity of the receptor, by FACS or single-cell microscopy, the use of transient transfection generates very rich datasets.

For simplicity, in this paper we select a narrow level of expression for which to report the activity – this is quite different from whole-cell Western blots, where the entire expression range is collapsed into a single measurement. Thus, in effect, our data are more akin to results obtained with stable transfection, with the advantage that we can readily choose the expression level for which we analyze the data. For the data shown in the paper, we chose an expression window just at the point where the surface expression of EGFR flattens with increasing total expression (for both wild type and mutant EGFR). This gives us readings at a level of expression where the signal to noise is high, but below a point where overexpression saturates the membrane trafficking machinery (this effect can be seen in the new Figure 8—figure supplement 2, which shows surface expression of wild-type and mutant EGFR).

Regarding mass spectrometry, we agree that this could be more quantitative, but the experiments that would be needed to acquire accurate and quantitative data would introduce a whole new set of experimental development that we feel is beyond the scope of the present work. We agree that this would be a fruitful avenue for future experimentation.

*Similarly, the arguments surrounding Figure 10 have to be clarified. What these data show is simply that loss of 'activator' function leads to loss of Y1173 phosphorylation. How does this translate into an argument that multimerization might be a mechanism for boosting phosphorylation of proximal and distal sites on the tail?*

Since loss of activator function leads to loss of 1173 phosphorylation, we reasoned that in a multimer, where the majority of kinases are “activators”, phosphorylation would be boosted. We have modified the text in paragraph ten, subheading “Functional role of multimerization in EGFR autophosphorylation and downstream signaling” of the revised manuscript to clarify this reasoning.

*6) Likewise does dimerization arm deletion increases two-step bleaching events in any specific way, or is this a sign of aggregation. Does this form still bind EGF the same?*

The nature of the two-step bleaching events we observed for EGFR and variants, in the absence of EGF, is a very interesting issue, and we would like to study this further in the future.

For the dimerization arm deletion construct, we observed increased two-step bleaching events (Figure 3), without seeing any sign of multistep bleaching events in the absence of EGF. When we made additional mutations at the asymmetric kinase dimer interface in this dimerization arm deletion construct, we observed a reduction of the two-step bleaching events. This suggests that these two-step bleaching events are unlikely to result from non-specific protein aggregation. We reasoned that the increased two-step bleaching events we observed for the dimerization arm deletion construct (Figure 3) and the other domain IV tether deletion construct (Figure 3) are a consequence of released autoinhibition, which converts the extracellular domain to a more extended conformation that does not impede dimerization. Note, however, that we have removed discussion of the nature of the ligand-independent dimerization from the revised manuscript.

We have not studied the binding property of the dimerization arm deletion construct. Since this matter is somewhat outside the main focus of the paper, we reserve this for future study. Mark Lemmon’s group has studied the effects of various EGFR extracellular domain mutations on EGF binding in the isolated extracellular domain, and they find a minimal effect due to the dimerization arm mutations (Dawson et al., MCB, 2005, PMID:16107719 and Dawson et al., Structure, 2007, PMID: 17697999).

*7) The specificity of the P-Tyr specific antibodies should be demonstrated using EGFR mutants Y955F, Y992F, Y1173F, Y1186F etc.*

We have studied this carefully and reported the results in a previous publication (Kovacs et al., MCB, 2015, PMID: 26124280). Based on this study we chose antibodies against Tyr 992 and Tyr 1173 for use in this paper, because they are the most specific anti-EGFR phosphotyrosine antibodies.

*8) Could the GFP fusion partner alter the association/dissociation of EGFR? This is especially important as it pertains to the formation of the higher order complexes. Indeed the authors point out that their previous work with GCN4 fusions led to some unexpected associations. Could an experiment or two with antibody (Fab) Q-dots with non-GFP fused receptors be helpful?*

Since antibodies bind to the surface of the extracellular domain, they may perturb the multimerization in ways that are difficult to predict. For this reason, we have avoided this approach. The FCCS experiments show that EGFP mutant used in these studies on its own does not dimerize to an appreciable extent.